# Variety in the USP deubiquitinase catalytic mechanism

Niels Keijzer[1], Anu Priyanka[1], Yvette Stijf-Bultsma[1], Alexander Fish[1], Malte Gersch[2,3], Titia K Sixma[1]

The ubiquitin-specific protease (USP) family of deubiquitinases (DUBs) controls cellular ubiquitin-dependent signaling events. This generates therapeutic potential, with active-site inhibitors in preclinical and clinical studies. Understanding of the USP active site is primarily guided by USP7 data, where the catalytic triad consists of cysteine, histidine, and a third residue (third critical residue), which polarizes the histidine through a hydrogen bond. A conserved aspartate (fourth critical residue) is directly adjacent to this third critical residue. Although both critical residues accommodate catalysis in USP2, these residues have not been comprehensively investigated in other USPs. Here, we quantitatively investigate their roles in five USPs. Although USP7 relies on the third critical residue for catalysis, this residue is dispensable in USP1, USP15, USP40, and USP48, where the fourth critical residue is vital instead. Furthermore, these residues vary in importance for nucleophilic attack. The diverging catalytic mechanisms of USP1 and USP7 are independent of substrate and retained in cells for USP1. This unexpected variety of catalytic mechanisms in this well-conserved protein family may generate opportunities for selective targeting of individual USPs.

## Introduction

Deubiquitinating enzymes (DUBs) are isopeptidases that remove ubiquitin from target substrates. This regulation of ubiquitin conjugation is essential, as it is involved in many different cellular pathways. Ubiquitination can cause changes in localization, activation, and signaling of the protein, or facilitate its degradation by the proteasome or lysosome (Hershko & Ciechanover, 1998). Ubiquitin itself can be modified on its seven lysines or its N-terminal methionine, thereby generating a variety of ubiquitin chain types, adding to the vast variety in signaling potential of ubiquitin (Komander & Rape, 2012). By removing ubiquitin or ubiquitin chains from target substrates, DUBs play a vital role in regulating pathways throughout the cell and inhibition of DUBs is therefore a viable therapeutic strategy. In fact, small molecule inhibitors for USP1 and USP30 are currently being explored in clinical studies against

cancer and renal disease, respectively (Cadzow et al, 2020; Tsefou et al, 2021).

Ubiquitin-specific proteases (USPs) form the largest known family of deubiquitinating enzymes (Nijman et al, 2005). They were originally discovered in yeast (termed UBPs) and were identified as cysteine proteases of the papain superfamily, because of their extremely conserved cysteine and histidine residues (Baker et al, 1992; Papa & Hochstrasser, 1993; Barrett & Rawlings, 1996). Together with a third catalytic residue, these form a catalytic triad similar to other proteases (Stroud, 1974; Storer & Ménard, 1994). To cleave the ubiquitin–substrate peptide bond, the catalytic cysteine acts as the nucleophilic agent and provides the reactive thiol group. The properties of this cysteine, enabling nucleophilic attack, are endowed by a histidine (Polgár, 2013). The third catalytic residue forms a hydrogen bond with this catalytic histidine, thereby polarizing it and allowing it to stabilize the catalytic cysteine (Vernet et al, 1995). After the nucleophilic attack, a tetrahedral thioester intermediate is generated, which is then hydrolyzed to regenerate the free enzyme. Transition states for the formation and hydrolysis of the thioester are stabilized by the oxyanion hole, an essential network of hydrogen bonds provided by neighboring residues (Ménard & Storer, 1992).

All USPs contain a conserved catalytic domain (~350 amino acids), generally decorated with additional domains that add to their extensive structural and functional variety (Komander et al, 2009; Ye et al, 2009). The conformation of the USP catalytic triad was first revealed when the structure of the USP7 catalytic domain was published (Hu et al, 2002). It was shown that USP7's catalytic triad consists of cysteine (C223), histidine (H464), and mutagenesis identified an aspartate (D481) as the third catalytic residue. Directly adjacent to this aspartate lies another aspartate, and the structure showed that this residue plays an essential role in stabilization of the tetrahedral intermediates via a nearby water molecule (Hu et al, 2002). In the context of the structural analysis of the catalytic domain of USP2, it was realized that either residue (N574, D575) is sufficient for catalytic activity, as mutations in either residue individually had no significant effect on catalysis (Zhang et al, 2011). Regardless of these findings in USP2, the assignment of the catalytic triad in other USPs has often been based on USP7, using sequence and structural alignments of its catalytic residues.

The third catalytic residue varies among USPs and can be either an aspartate, asparagine, or serine. The succeeding residue is more

[1]Division of Biochemistry and Oncode Institute, Netherlands Cancer Institute, Amsterdam, Netherlands   [2]Max Planck Institute of Molecular Physiology, Chemical Genomics Centre, Dortmund, Germany   [3]Department of Chemistry and Chemical Biology, TU Dortmund University, Dortmund, Germany

Correspondence: t.sixma@nki.nl

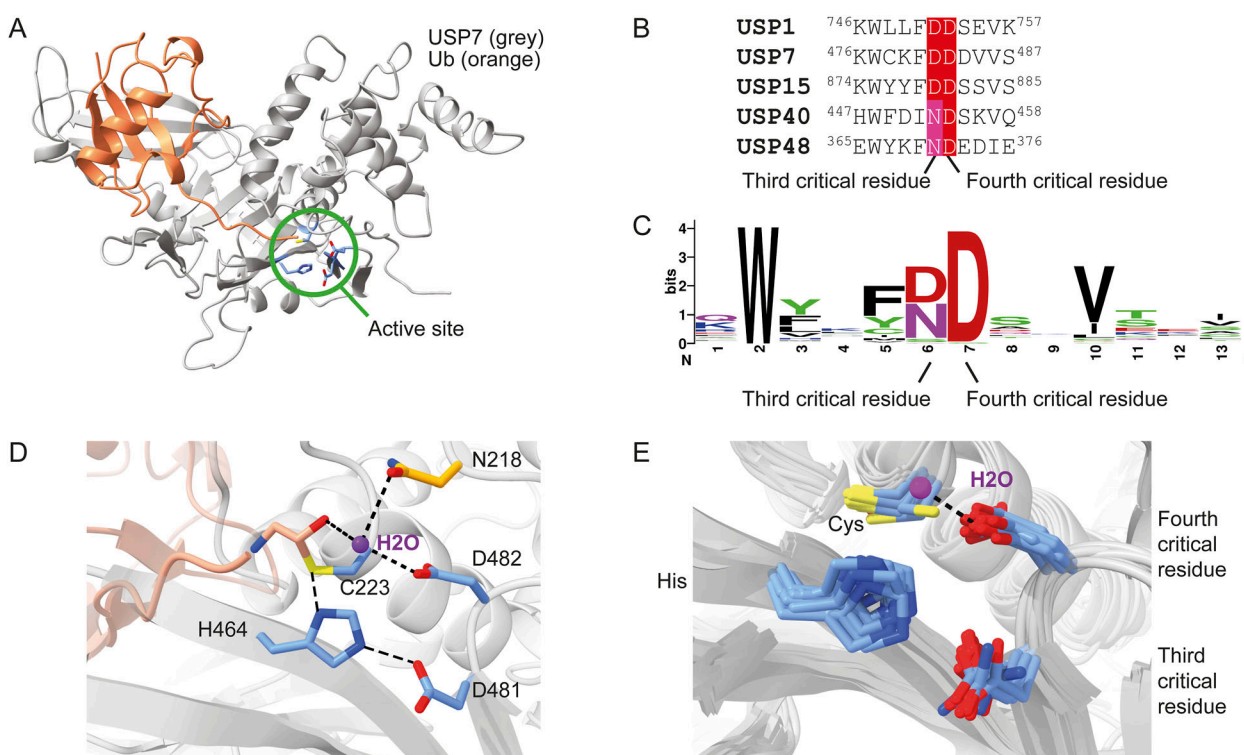

**Figure 1. Structural and sequence alignments of ubiquitin-specific proteases (USPs) show that both critical residues are positioned close to the catalytic histidine.**
**(A)** Crystal structure of USP7 (gray) bound to ubiquitin-aldehyde (orange) (PDB: 1NBF). The location of the active site is highlighted (green), and catalytic residues with the adjacent extremely conserved aspartate are shown in blue. **(B)** Sequence alignment of the five human USPs relevant for this study. **(C)** Sequence logo of the 56-known human USPs; the canonical third catalytic residue (6) is referred to as the third critical residue. The highly conserved adjacent residue (7) is referred to as the fourth critical residue. The full sequence alignment is shown in Fig S1B. **(D)** Canonical catalytic mechanism shown by USP7 in complex with ubiquitin-aldehyde (1NBF, USP7: gray; ubiquitin-aldehyde: pink). Catalytic cysteine (C223: first critical residue), histidine (H464: second critical residue), the canonical third catalytic residue (D481: third critical residue), and adjacent highly conserved aspartate (D482: fourth critical residue) are shown in blue. Hydrogen bonds (<3.5 Å) are shown as black dashed lines. The oxygen atom of the third critical residue (D481) forms a hydrogen bond with the nitrogen (Nε) on catalytic histidine (H464), which allows histidine to activate cysteine for the nucleophilic attack. A water molecule (purple) is held in place by the fourth critical residue and an asparagine (N218, orange). This water molecule acts as a member of the oxyanion hole to stabilize ubiquitin (pink). **(E)** Superposition of available USP structures (Table 1) with aligned catalytic triads shows subtle variations in the positioning of the residues. As the structure of USP35 has a cysteine-to-serine mutation, this residue is not shown. Hydrogen bonding of the fourth critical residue with water molecule is only present in USP7.

highly conserved, as it is predominantly an aspartate in human USPs. However, despite this conservation, the importance and precise role of the third and fourth critical residues have not been assessed among the wider USP family. In contrast, based on structural analysis alone, a handful of papers assign the fourth critical residue as the catalytic residue, but do not verify it by mutagenesis (Pereira et al, 2015; Yin et al, 2015; Leznicki et al, 2018), and it is not clear whether there exists a unified mechanism for this DUB family.

Several USPs harbor a misaligned catalytic triad in their apo-structures, which was first shown in USP7 (Hu et al, 2002). Upon ubiquitin binding, the switching loop (SL) of USP7 changes conformation, which causes a rearrangement of the catalytic triad into its catalytically competent conformation (Faesen et al, 2011; Kim et al, 2016). USP40 has a USP7-like activation mechanism, implying a similar inactive state (Kim R, personal communication). The structure of the USP15 catalytic domain (Ward et al, 2018) also harbors a misaligned catalytic triad, but unlike USP7, USP15 does not require ubiquitin binding in order to realign and instead undergoes conformational changes before ubiquitin binding (Priyanka et al, 2022). Although the SL is structurally well conserved among USPs, such a conformational change has only been observed in USP7, USP15, and USP34 (Xu et al,

2022). The apo-structure of USP4, which shares domain structure with USP15, did not show a misaligned catalytic triad (Clerici et al, 2014).

Here, we revisit the catalytic mechanism assignment of USP DUBs. We investigate five diverse USPs (USP1, USP7, USP15, USP40, and USP48) to assess the importance of the third catalytic residue and the adjacent aspartate for catalysis, which we here term the third and the fourth critical residue, respectively (with catalytic cysteine and histidine being the first and second critical residues).

We reveal that instead of the canonical third catalytic residue, USP1, USP15, USP40, and USP48 rely on the fourth critical residue for catalysis. USP15 does not require the canonical third catalytic residue for catalysis, as an alanine substitution mutant still behaves like the WT enzyme. USP1, USP40, and USP48 only suffer from a small decrease in activity when the third critical residue is lost. Only USP7 is rendered catalytically dead when its third catalytic residue is mutated. In each USP, we find that the critical residue most important for full catalysis is also the residue that is responsible for accommodating the nucleophilic attack, except for USP15, where this can be accomplished by either residue. We verify the importance of these critical residues by analyzing the effect of these mutations in cells for USP1. Our results demonstrate that using structural and sequence alignments alone does not predict whether the

third catalytic residue or the adjacent aspartate residue is more essential and that a surprising degree of plasticity exists between the catalytic components of USPs. This finding has important implications for the dissection of catalytic mechanisms of other USPs and suggests that opportunities for selective targeting could exist.

# Results

## Positioning of catalytic residues in the catalytic cleft is highly similar in different USPs

Directly adjacent to the canonical third catalytic residue (third critical residue) is an even more highly conserved aspartate (the fourth critical residue) (Figs 1A–D and S1A). This aspartate is present in all USPs except CYLD and USP50. The latter misses the third critical residue as well and therefore may be inactive.

It was previously shown that there are no structural differences in the positioning of the catalytic triad and the fourth critical residue between USP2 and USP7, despite their third and fourth critical residues behaving differently (Zhang et al, 2011). We superimposed the currently available crystal structures of USP catalytic domains (Table 1, Fig 1E) and also found only minor differences in the positioning of these two adjacent residues. Moreover, the water, which interacts with the fourth critical residue, required for oxyanion hole formation, is exclusively found in USP7, despite the fact that some of these structures should have sufficient resolution to identify water molecules (Table 1).

We set out to study the role of these adjacent critical residues in a selection of diverse USPs (USP1, USP7, USP15, USP40, and USP48). These USPs vary considerably in domain architecture and allosteric regulation, and therefore represent different aspects of the USP family, known for its structural variety and modular architecture (Nijman et al, 2005). USP1, USP7, and USP15 harbor two aspartates as the third and the fourth critical residue, whereas USP40 and USP48 harbor an asparagine and aspartate as the third and the fourth critical residue respectively, allowing us to examine the importance of a negative charge in the position of the third critical residue. Although the structures of USP40 and USP48 have not been solved, they contain the conserved USP catalytic domain and AlphaFold predictions for USP40 (UniProt: Q9NVE5) and USP48 (UniProt: Q86UV5) do not suggest major changes in their catalytic domains. To allow accurate functional assessment in a side-by-side comparison, we generated the following mutations: USP1 (Res3: D751A; Res4: D752A), USP7 (Res3: D481A; Res4: D482A), USP40 (Res3: N452A; Res4: D453A), USP48 (Res3: N370A, Res4: D371A), and the D1D2 catalytic core of USP15 (Ward et al, 2018; Priyanka et al, 2022), USP15D1D2 (Res3: D879A; Res4: D880A).

As a quality control, we assessed potential misfolding or instability of all mutants tested using a thermal stability assay (Fig S1B, Table S1). Comparing the individual critical residue alanine mutants with their WT counterpart, we observed only minor differences in thermal stability. Although both mutants of USP15 have a decreased thermal stability compared with USP15wt, these variants retain stability until 50°C, indicating that they are still well-folded and suitable for kinetic assays at RT.

**Table 1. List of ubiquitin-specific proteases used in superposition, corresponding PDB identifiers, state of the protein, and whether it is bound to a ubiquitin or ubiquitin-like protein, full length (FL), catalytic domain (CD), and cofactors and resolution.**

|        | PDB-ID | Ubiquitin(-like protein) | Domains | Resolution |
|--------|--------|--------------------------|---------|------------|
| USP1   | 7AY2   | Ub$^{Propargyl}$         | CD + UAF1 | 3.2 Å    |
| USP2   | 2HD5   | Ub$^{wt}$                | CD      | 1.85 Å     |
| USP4   | 2Y6E   | Apo-form                 | CD (D1D2) | 2.4 Å    |
| USP7   | 1NBF   | Ub$^{Aldehyde}$          | CD      | 2.3 Å      |
| USP8   | 2GFO   | Apo-form                 | CD      | 2.0 Å      |
| USP9X  | 5WCH   | Apo-form                 | CD      | 2.5 Å      |
| USP12  | 5L8W   | Ub$^{Bromoethylamine}$   | FL + UAF1 | 2.8 A    |
| USP14  | 2AYO   | Ub$^{Aldehyde}$          | CD      | 3.5 Å      |
| USP15  | 7R2G   | Active form              | CD (D1D2) | 1.98 Å   |
| USP18  | 5CHV   | ISG15                    | CD      | 3.0 Å      |
| USP21  | 2Y5B   | Linear di-Ub$^{Aldehyde}$ | CD     | 2.7 Å      |
| USP25  | 6HEI   | Ub$^{Propargyl}$         | CD      | 1.64 Å     |
| USP28  | 6HEK   | Ub$^{Propargyl}$         | CD      | 3.03 Å     |
| USP30  | 5OHK   | Ub$^{Propargyl}$         | CD      | 2.34 Å     |
| USP35  | 5TXK   | Ub$^{wt}$                | CD      | 1.84 Å     |
| USP46  | 5L8H   | Ub$^{vinylmethylester}$  | Full length | 1.85 Å |

## The third critical residue is dispensable in USP1/UAF1, USP15, USP40, and USP48

To measure the catalytic activities of WT and mutants of each USP, we followed enzyme activity using fluorogenic substrates, ubiquitin-rhodamine (Ub$^{Rho}$) or ubiquitin-AMC (Ub$^{AMC}$), at varying concentrations (Figs 2A and S2A and B). We then studied the kinetics of these variants by Michaelis–Menten analysis (Table S2) and compared their catalytic efficiencies to assess the relative importance of both critical residues (Table 2). Surprisingly, our experiments reveal that USP1/UAF1, USP15, USP40, and USP48 do not rely on the third critical residue, canonically considered the third catalytic residue, and instead were rendered catalytically inactive only when their fourth critical residue is mutated.

Examination of the Michaelis–Menten analysis shows that USP1, USP15, USP40, and USP48 are all still catalytically competent upon loss of their third critical residue (Fig 2B, Table 2). The loss of the third critical residue leads to minor decreases in their catalytic efficiency, to varying extents. In USP15, we can observe that USP15$^{D879A}$ does not decrease its catalytic efficiency at all compared with USP15$^{wt}$. USP1$^{D751A}$ only suffers from a minor 1.4-fold decrease in catalytic efficiency compared with USP1$^{wt}$, and USP48$^{N370A}$ suffers a twofold decrease compared with USP48$^{wt}$. This effect is slightly bigger in USP40$^{N452A}$, which decreases its catalytic efficiency fourfold. Interestingly, of the five USPs we tested, USP7 was the only USP where we observe a major effect on catalytic efficiency. Mutagenesis of the third critical residue (USP7$^{D481A}$) leads to a large (26-fold) decrease in catalytic efficiency ($k_{cat}/K_M$) compared with USP7$^{wt}$, rendering USP7$^{D481A}$ mostly inactive, which agrees with existing literature on USP7 in a gel-based activity assay (Hu et al, 2002).

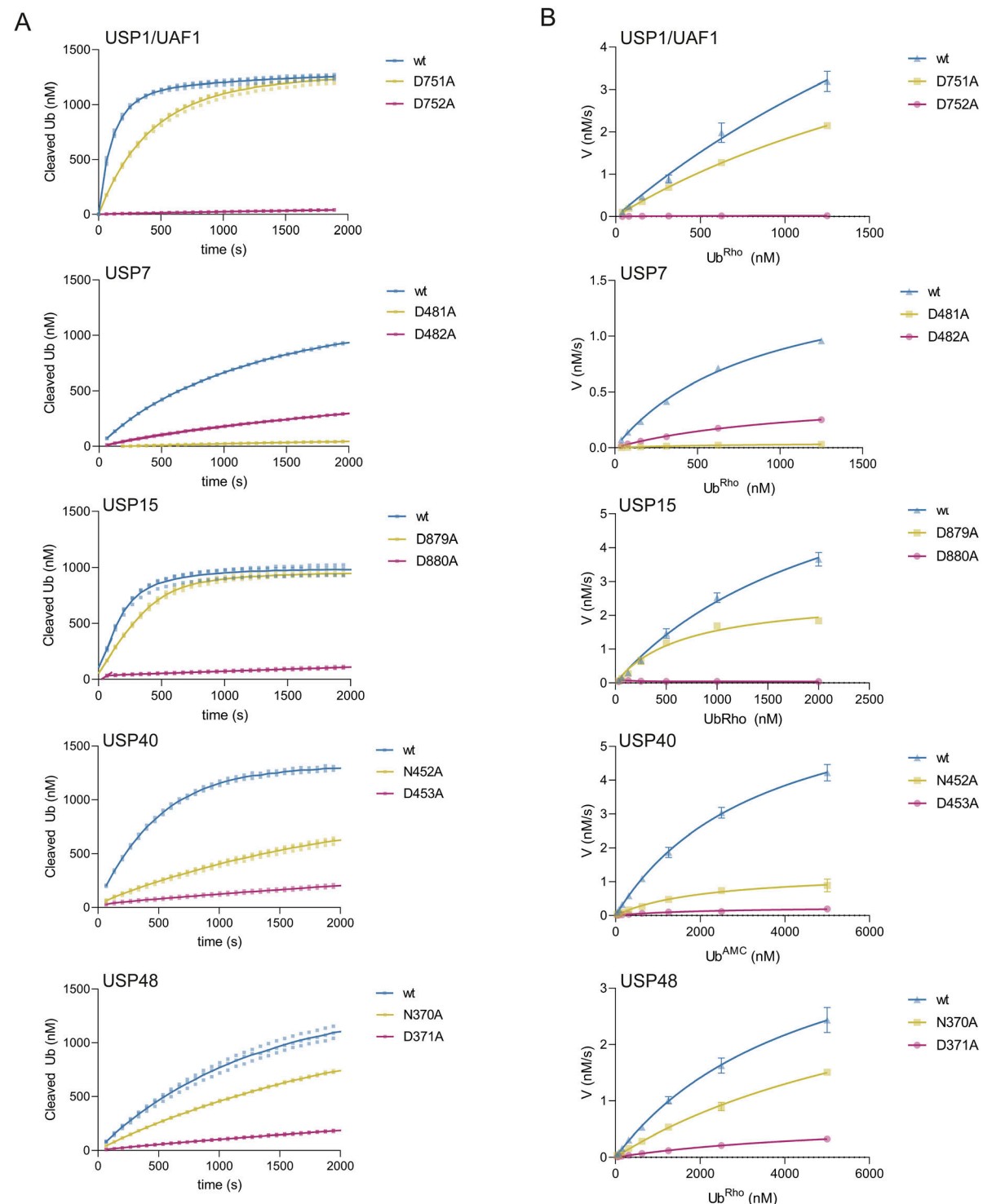

**Figure 2. USP1, USP15, USP40, and USP48 do not rely on the canonical third critical residue.**
**(A)** Enzyme activity assays of USPs on a minimal substrate (Ub$^{Rho}$ for USP1/UAF1, USP7, USP15, and USP48; Ub$^{AMC}$ for USP40). For each USP, we show a single enzyme concentration against a single concentration of substrate. WT (blue) and mutation of the third critical residue (yellow) and the fourth critical residue (purple) are shown. Assays with a full range of substrate concentrations are shown in Fig S2A and B (n = 2 biological replicates; n = 3 technical replicates). **(B)** Michaelis–Menten kinetics of USP variants, based on activity assays shown in Fig S2A and B.

**Table 2. Catalytic efficiencies ($k_{cat}/K_M$) of USPs and their critical residue mutants on a minimal substrate.**

| | WT ($s^{-1}$ $\mu M^{-1}$) | Third critical residue ($s^{-1}$ $\mu M^{-1}$) | Fourth critical residue ($s^{-1}$ $\mu M^{-1}$) |
|---|---|---|---|
| USP1/UAF1 | wt 0.35 (±0.17) | D751A 0.25 (±0.015) | D752A 0.005 (±0.00036) |
| USP7 | wt 1.87 (±0.22) | D481A 0.07 (±0.008) | D482A 0.43 (±0.047) |
| USP15$^{D1D2}$ | wt 0.35 (±0.074) | D879A 0.38 (±0.102) | D880A < 0.0001 |
| USP40 | wt 0.21 (±0.0065) | N452A 0.06 (±0.0073) | D453A 0.01 (±0.0035) |
| USP48 | wt 0.02 (±0.0009) | N370A 0.01 (±0.001) | D371A 0.002 (±0.0002) |

The full Michaelis–Menten analysis is shown in Table S2. Analogous KinTek verification of the Michaelis–Menten analysis is shown in Fig S3 and Table S3.

### The fourth critical residue is essential for catalysis in USP1, USP15, USP40, and USP48

In contrast, our analysis revealed that the other USPs suffer a significant decrease in catalytic efficiency when their fourth critical residue is mutated, which results in USP1$^{D752A}$, USP15$^{D880A}$, USP40$^{D453A}$, and USP48$^{D371A}$ showing virtually no activity. Catalytic efficiency of USP1$^{D752A}$ is 86-fold lower compared with USP1$^{wt}$. In USP15$^{D880A}$, we were not able to measure a signal of released fluorescent rhodamine at all, indicating that there is no activity left. Mutation of D453A in USP40, that is, loss of the fourth critical residue, causes a 20-fold decrease in catalytic efficiency compared with USP40$^{wt}$. USP48$^{D371A}$ shows a 10-fold decrease in catalytic efficiency compared with USP48$^{wt}$, which itself has a relatively low catalytic efficiency on Ub$^{Rho}$, even while using a higher enzyme concentration (100 nM). Still, this 10-fold decrease renders USP48$^{D371A}$ almost catalytically dead.

We find that the fourth critical residue, presumably its role as oxyanion for intermediate stabilization, is only of minor importance in USP7, as seen by a small (fourfold) decrease in catalytic efficiency of USP7$^{D482A}$ compared with USP7$^{wt}$, a smaller decrease than previously seen for this mutant (Hu et al, 2002). These results do confirm that the third critical residue in USP7 makes up the catalytic triad. Our Michaelis–Menten analysis therefore implies that the role of the fourth critical residue in stabilization of the tetrahedral intermediate in USP7 is less important than was first thought.

Alanine mutations themselves leave an open space and could thereby affect the local environment in the active site. We therefore generated asparagine mutations of the third and fourth critical residues in USP1 to account for this effect (Fig S2B and C). Results of careful kinetic assays show that the fourth critical residue mutant USP1$^{D752N}$ barely shows any activity and the activity of USP1$^{D751N}$ is reduced compared with USP1$^{wt}$. In addition, the activity of USP1$^{D751N}$ appears to be lower compared with USP1$^{D751A}$. This suggests that the precise residue found in the third critical residue is important for catalysis in USPs, which agrees with earlier findings on a serine in this position (Gersch et al, 2017).

Our findings demonstrate that for most of the tested USPs, the third critical residue appeared dispensable and that there exists a variety in catalytic importance of the two critical residues between different USPs. Interestingly, with the exception of USP7, the fourth critical residue is the essential residue for catalysis in most USPs tested here and is able to accommodate full catalysis in the absence of the third critical residue.

### Activity on natural substrates confirms different catalytic triad compositions

To validate that USP1/UAF1 can really function without the canonical third critical residue, we compared the activity of USP1/UAF1 and USP7 on a natural substrate. Deubiquitination activity was tested on mono-ubiquitinated PCNA (PCNA-Ub), a well-known substrate of USP1/UAF1 (Huang et al, 2006) and a potential substrate of USP7 (Kashiwaba et al, 2015). We found that USP1/UAF1, USP7, and their mutants display the same relative activity toward PCNA-Ub as was observed for the minimal substrate. As expected, USP1$^{wt}$/UAF1 and USP7$^{wt}$ were able to cleave PCNA-Ub (Fig 3A and B). The third critical residue mutant USP1$^{D751A}$/UAF1 causes a minor decrease in cleavage of PCNA-Ub compared with USP1$^{wt}$/UAF1. We find that just like their activity on Ub$^{Rho}$, USP1$^{D752A}$ (fourth critical residue) and USP7$^{D481A}$ (third critical residue) are unable to cleave PCNA-Ub, with USP7$^{D481A}$ still retaining minimal activity. Mutating the fourth critical residue in USP7$^{D482A}$ displays only a minor decrease in activity compared with USP7$^{wt}$, although this could be due to a difference in the enzyme/substrate ratio. Taken together, the different relative importance of the first and second residue is retained on different substrates.

### USP1 relies on the fourth critical residue to process PCNA-Ub in cells

We wondered whether the importance of the fourth critical residue for catalysis in USP1 also holds up in a cellular context. Using doxycycline-inducible lentiviral expression in mammalian (RPE1) cells, we complemented a USP1 knockout cell line with full-length USP1$^{wt}$, USP1$^{D751A}$, and USP1$^{D752A}$ to investigate whether their activity is comparable to that seen on Ub$^{Rho}$ and to in vitro activity on PCNA-Ub. As a control, cells were complemented with catalytically dead USP1$^{C90R}$ (Morrow et al, 2018) to validate our experimental setup. Between single clones of our USP1 knockout cell line, we observed variable levels of basal PCNA-Ub even before inducing expression with doxycycline (Fig S4). We therefore calculated the ratio between PCNA-Ub levels before and after doxycycline induction within single clones, and provide ratios averaged over multiple clones. Expressing USP1$^{wt}$ caused a decrease in PCNA-Ub levels, whereas USP1$^{C90R}$ expression did not affect PCNA ubiquitination levels, confirming that our experimental setup was functioning (Fig 3C).

Introducing USP1$^{D751A}$ and USP1$^{D752A}$ in the USP1 knockout cells confirmed that also in cells, mutation of the third critical residue (USP1$^{D751A}$) does not affect its catalytic activity. Instead, the fourth critical residue mutant (USP1$^{D752A}$) causes loss of almost all activity, comparable to USP$^{C90R}$, as judged by unchanged or accumulating levels of PCNA-Ub after doxycycline induction. Taken together, these experiments validate the

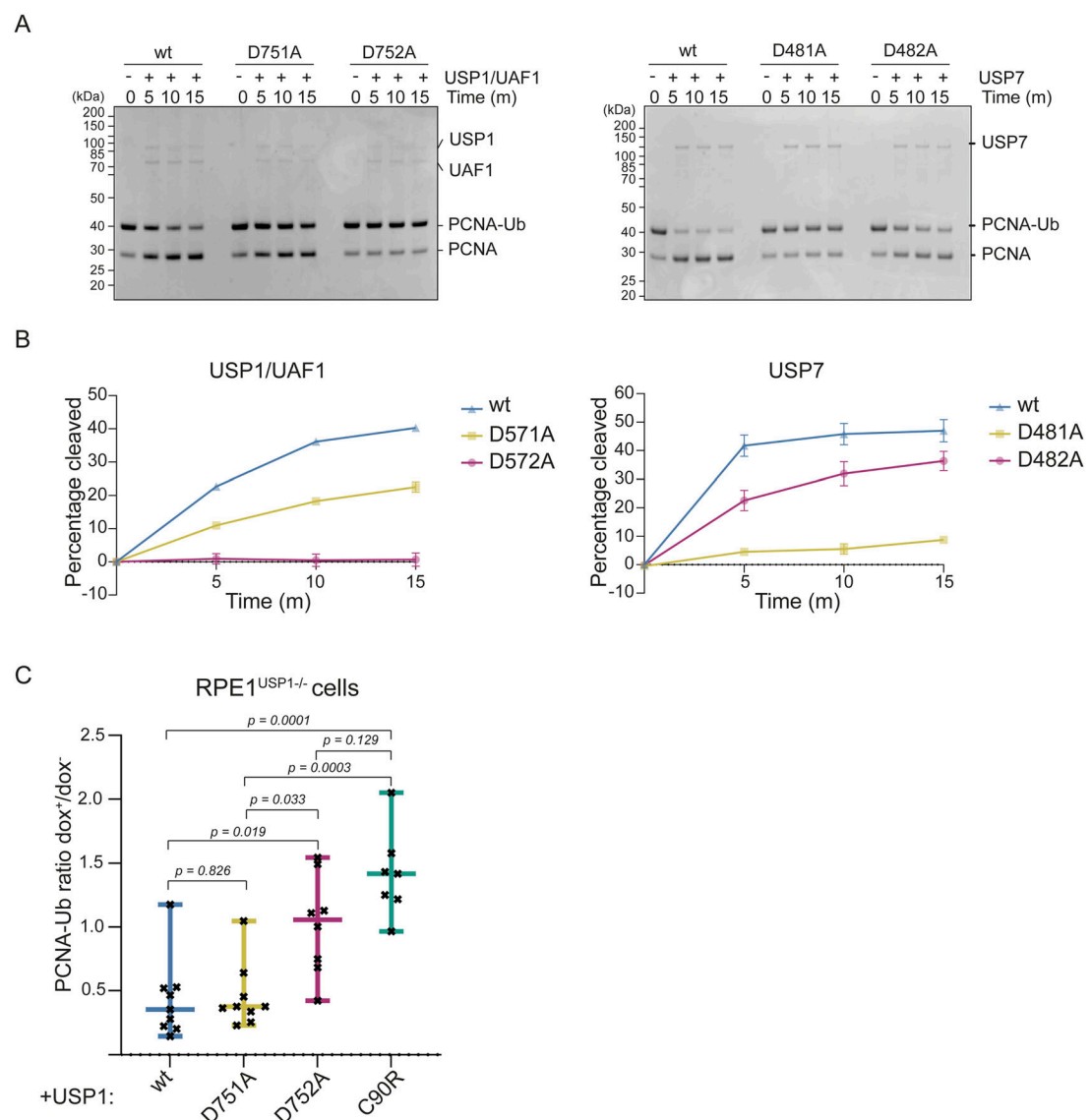

**Figure 3. Varying use of critical residues in USP1 and USP7 is retained on a natural substrate and confirmed for USP1 in cells.**
**(A)** In vitro deubiquitination assay of USP1/UAF1 and USP7 on a natural substrate (PCNA-Ub) comparing the importance of both critical residues. Results confirm that the third critical residue is more critical for USP7, and the fourth critical residue is more critical for USP1/UAF1. **(B)** Quantification of gel-based activity assays. USP1[D752A] and USP7[D481A] are catalytically incompetent, whereas USP1[D751A] and USP7[D482A] are still able to cleave PCNA-Ub. **(C)** Quantification of PCNA deubiquitination in USP1 RPE1 knockout cell lines complemented with USP1[wt] and USP1[D751A], USP1[D752A], and USP1[C90R] (Fig S4). Cell lysates were stained using antibodies for PCNA-Ub, PCNA, and USP1. Levels of PCNA-Ub were quantified before (dox[−]) and after (dox[+]) doxycycline induction to determine the ratio between PCNA-Ub levels. *P*-values (see the Materials and Methods section) are shown, and results confirm that USP1[D751A] behaves like WT and that USP1[D752A] significantly loses activity.
Source data are available for this figure.

importance of the fourth critical residue for cellular substrate processing. Moreover, they suggest that the outcome of in vitro analyses of USP catalytic mechanisms translate into relevant cellular substrate turnover.

**Different catalytic triad compositions are not affected by pH**

Next, we decided to assess the differential importance of both residues in more detail, to define their precise roles during substrate processing. We examined the effect of buffer pH on the activity of different mutants, aiming at elevated pH because this could promote deprotonation of the catalytic cysteine. This in turn would make the

first step of the catalytic cycle less dependent on the other two residues of the catalytic triad. In addition, USP7[FL] and USP7[CD] were previously shown to have elevated activity at higher pH (Faesen et al, 2011). We performed the DUB activity assays at pH 7.0, pH 8.0, and pH 9.0 against a single concentration of minimal substrate (Fig S5A). Results show that the relative importance of the critical residues in USP1, USP7, USP15, and USP48 is not influenced by a change in pH. Although these USPs are more active at pH 8.0 and pH 9.0, their diverging preference for different critical residues remains.

In USP40, the mutants appear to be more pH-sensitive than USP40[wt], with USP40[N452A] and USP40[D453A] having lower activity at pH

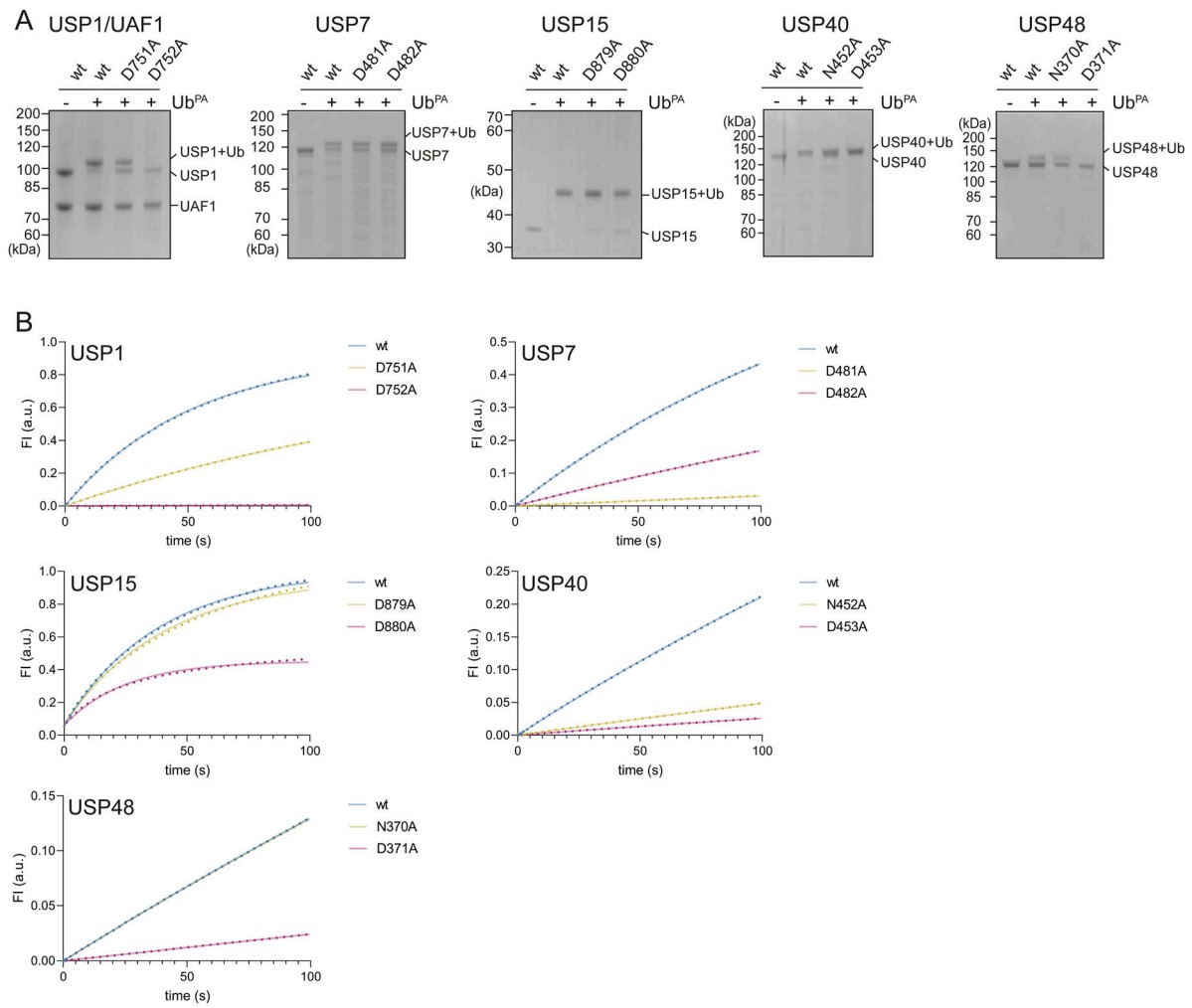

**Figure 4. Variation in the ability of ubiquitin-specific protease (USP) critical residue mutants to successfully facilitate a nucleophilic attack.**
**(A)** Analysis of the ability of USPs and their critical residue mutants to successfully conjugate to ubiquitin-propargyl (Ub$^{PA}$). Only the fourth critical residue is able to accommodate conjugation to Ub$^{PA}$ in USP1/UAF1 and USP48, but both critical residues can accommodate Ub$^{PA}$ conjugation in USP7, USP15, and USP40. **(B)** Stopped-flow analysis of initial rates of USP variants. USP15's fourth critical residue mutants show high initial activity but reach a lower plateau compared with USP15$^{wt}$ and the third critical residue mutant. USP1, USP7, USP40, and USP48 behave as was observed in the Michaelis–Menten analysis. USP1's fourth critical residue mutant does not appear to be active at all. Although USP48$^{D371A}$ shows minimal activity, a higher enzyme (50 nM) and substrate concentration (500 nM) were used to overcome USP48's low activity on a minimal substrate.
Source data are available for this figure.

7.0 and pH 9.0, but a higher activity at pH 8.0. In addition, at a higher pH (8.0 and 9.0), the activity of USP40$^{N452A}$ and USP40$^{D453A}$ becomes more similar, whereas at a lower pH (7.0), USP40$^{N452A}$ retains more activity than USP40$^{D453A}$, and this is also seen when comparing pH 8.0 to their activity to the previously tested pH 7.5 (Fig S5B). Altogether, the pH experiments suggest that the fourth critical residue of USP1, USP15, USP40, and USP48 and the third critical residue of USP7 are involved in cysteine deprotonation regardless of pH.

### The third and fourth critical residues play varying roles in accommodating a nucleophilic attack

Cleaving a ubiquitin–substrate bond is not a single event, but instead a series of events that culminates in the release of ubiquitin and the substrate. The two adjacent critical residues play different

roles in this process, either polarizing histidine, which in turn allows cysteine to attack the ubiquitin–substrate linkage, or resolving the tetrahedral intermediates formed after the nucleophilic attack. We used the probe ubiquitin-propargyl (Ub$^{PA}$) to directly analyze the first step, the nucleophilic attack, and therefore separate the need to resolve the tetrahedral intermediates to complete the catalytic cycle. We thus compared the ability of the different USP variants to react with Ub$^{PA}$ (Fig 4A), allowing us to specifically investigate which critical residue is necessary to polarize the catalytic histidine.

Interestingly, the catalytic cysteine in USP1 and USP48 is unable to conjugate to Ub$^{PA}$ when the fourth critical residue is lacking (USP1$^{D752A}$ and USP48$^{D371A}$), but is able to conjugate when the third critical residue is mutated (USP1$^{D751A}$ and USP48$^{N370A}$). This is noteworthy as the propargylamine warhead does not feature the carbonyl of Gly76, which would be stabilized by the oxyanion hole

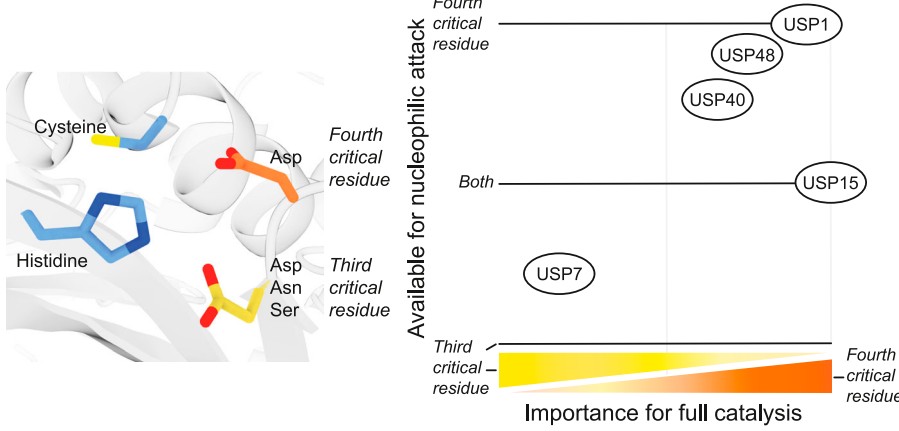

**Figure 5. Critical residues perform different functions in different ubiquitin-specific proteases (USPs).**
(Left panel) Depiction of the catalytic residues. Catalytic cysteine and histidine are shown in blue. The third and fourth critical residues are shown in yellow and orange, respectively. (Right panel) Ability of the critical residue to accommodate the nucleophilic attack is shown on the y-axis, and the importance of the critical residue for full catalysis is shown on the x-axis. These five USPs share the conserved USP catalytic domain but vary in domain architecture and allosteric regulation, and therefore represent a part of the diversity found in the USP family. In USP1, USP40, and USP48, only the fourth critical residue is able to accommodate an efficient nucleophilic attack. Both critical residues are able to polarize the catalytic histidine in USP15. In USP7, only the third critical residue can accommodate an efficient nucleophilic attack. After the nucleophilic attack, several tetrahedral intermediates need to be resolved in order to complete the catalytic cycle. In USP15, only the fourth critical residue is required to complete the catalytic cycle on its own, although both residues are able to polarize histidine. USP15 does not require the third critical residue to perform full catalysis. USP1, USP40, and USP48 rely mostly on the fourth critical residue, and the third critical residue only plays a minor role. USP7 primarily relies on the third critical residue and requires a minor contribution of the fourth critical residue.

residues. This suggests that in these two USPs, the fourth critical residue may be able to polarize the catalytic histidine and thereby accommodate the nucleophilic attack. In contrast, all USP7, USP15, and USP40 variants are able to conjugate to Ub$^{PA}$.

Our previous experiments have shown that in these three USPs, alanine mutations of the third critical residue (USP7$^{D481A}$) or the fourth critical residue (USP15$^{D880A}$, USP40$^{D453A}$) result in extremely low catalytically efficiencies (USP7 and USP40), or render it catalytically dead (USP15). Because these mutants are able to react with Ub$^{PA}$, we wondered whether they are able to perform an initial nucleophilic attack, but are unable to resolve the tetrahedral intermediates, rendering the enzymes inactive. Because our minimal substrate activity assays (Figs 2A and S2A and B) do not allow for detection of the first turnover, we performed stopped-flow analysis to gain insights into this initial phase of the reaction (Fig 4B).

USP15 is a fast enzyme and eventually depletes all of the substrate, which can be observed by the USP15$^{wt}$ and USP15$^{D879A}$ variants reaching a plateau. However, the USP15$^{D880A}$ mutant reaches a plateau lower than the other two variants, which indicates that it is unable to cleave all of the substrate. This suggests inactivation of USP15$^{D880A}$ and is consistent with it performing a single nucleophilic attack. Furthermore, the detection of fluorescence indicates that USP15$^{D880A}$ releases rhodamine, but not ubiquitin, by being unable to resolve its tetrahedral intermediates. Because of time resolution limitations in our plate reader assay, we were unable to detect the initial phase of the reaction during the minimal substrate activity assay, which caused this mutant to appear catalytically dead (Figs 2 and S2A and B).

This effect is not seen in the other USPs: USP7$^{D481A}$, USP40$^{D453A}$, and USP48$^{D371A}$ show very little activity against the minimal substrate in the stopped-flow setup, and USP1$^{D752A}$ does not appear to be active at all. This is in line with our Michaelis–Menten analysis, where USP1$^{D752A}$ and USP48$^{D371A}$ display the lowest catalytic efficiencies. We therefore tested whether the fourth critical residue mutants of USP1 and USP48 show any signs of Ub$^{PA}$ conjugation after a longer duration (60 min) (Fig S6). Even after longer duration,

USP1$^{D752A}$ is unable to conjugate to Ub$^{PA}$. In USP48$^{D371A}$, on the contrary, a minor amount is able to conjugate to Ub$^{PA}$ after 60 min.

In conclusion, the ability of USP48's fourth critical residue mutant to successfully perform a nucleophilic attack is extremely low and USP1's fourth critical residue mutant has completely lost this ability. On the contrary, the third critical residue mutant of USP7 and the fourth critical residue of USP40 can accommodate a nucleophilic attack, albeit with low efficiency. In contrast, USP15 does efficiently perform a nucleophilic attack using either critical residue. The fourth critical residue mutant of USP15 is, however, unable to perform full catalysis, which results in inactivation after the first turnover. Taken together, these data show that the two critical residues play different roles in accommodating a nucleophilic attack among these USPs and, thus, add another layer of variability in roles of these critical residues in the active site of USP DUBs.

# Discussion

In this study, we investigated the catalytic mechanism employed by USP DUBs and showed how the third and fourth critical residues contribute to catalysis with surprising variety. By combining data from nucleophilic attack and stopped-flow analysis (Fig 4) with those from the Michaelis–Menten analysis (Fig 2B, Table 2), we can generate a schematic figure that provides an overview of the roles of these residues among USPs (Fig 5). The two axes denote the potential of individual critical residue mutants to perform the nucleophilic attack (y-axis) and their importance for full catalysis (x-axis).

Most USPs tested here rely mostly on their fourth critical residue for catalysis (USP1, USP15, USP40, and USP48), some of which are virtually inactive without it. The third critical residue (the canonical third catalytic residue) only plays a minor role in full catalysis in USP1, USP40, and USP48, nor does it appear to play a significant role in performing the nucleophilic attack. Instead, these three USPs rely

on the fourth critical residue to perform the nucleophilic attack. USP15 makes up a different subgroup because it does not appear to require any involvement of the third critical residue to reach WT-like catalytic competence. Unlike the other USPs tested here, both its critical residues are able to efficiently facilitate the nucleophilic attack. Without the fourth critical residue, USP15 is rendered inactive after performing the nucleophilic attack, possibly as a result of its inability to resolve tetrahedral intermediates and release cleaved ubiquitin. Only USP7 lies on the other end of the spectrum and relies on the third critical residue for full catalysis. USP7 requires only a minor contribution of the fourth critical residue, which was previously shown to be due to its involvement in oxyanion stabilization (Hu et al, 2002).

These data demonstrate that the classical USP catalytic mechanism, as deduced from USP7 (Hu et al, 2002) or the alternative mechanism seen in USP2 (Zhang et al, 2011), does not apply to the USPs tested here. In fact, there is a remarkable variability in catalytic mechanisms among USPs as all five USPs tested here make different uses of the two critical residues for catalysis.

Although alanine mutations leave open an empty space, or take away the negative charge whenever an aspartate is mutated, mutating both critical residues to asparagine in USP1 did not alleviate the decrease in catalytic competence. In addition, all single critical residue mutants remained stable and some mutants retaining most of their catalytic competence suggest that these enzymes still function properly.

Previous research has shown that there are no discernable structural differences in the active sites of USP2 and USP7, which would explain their diverging mechanisms (Zhang et al, 2011). The structural alignment of USPs for which structures are available demonstrates that despite their structural variety, the positioning of the four critical residues remains highly conserved. The multiple sequence alignment too does not reveal obvious differences in the active site between USPs. Because of this, we propose that these alignments should be used with caution when predicting the importance of the third and fourth critical residues.

Most of the USPs tested here can complete the entire catalytic cycle without the canonical third catalytic residue (third critical residue), and we verified the activity of an active site without this residue in a cellular context. This indicates that these USPs are still able to deprotonate the catalytic cysteine to allow for the attack on the ubiquitin tail and subsequently resolve the tetrahedral intermediates and release of the product and substrate. It could be that in these USPs, hydrolase activity by a Cys-His dyad as is commonly observed in various other proteases is sufficient in the context of strong substrate activation by the oxyanion hole formation by the fourth critical residue and possibly other surrounding residues.

In our data, USP15 is unique, because it stops after a single turnover, when the fourth critical residue is absent. Its misaligned catalytic triad (Ward et al, 2018; Priyanka et al, 2022) suggests an intrinsic flexibility in the active site, which could allow for a rearrangement that allows also the fourth critical residue to hydrogen bond with the catalytic histidine. USP7 (Hu et al, 2002) and most likely USP40 (Kim R, personal communication) also have misaligned catalytic triads (Faesen et al, 2011; Kim R, personal communication). This structural plasticity could explain why mutations in one critical residue leave more residual activity compared with USP1 and USP48. However, an arrangement where the fourth critical residue forms a hydrogen bond with catalytic histidine has not been observed in any USP crystal structure and considering the relative arrangement would require major structural plasticity.

The third critical residue varying between aspartate, asparagine, and serine is an important feature of USPs, and the presence of a negative charge cannot directly be linked to the importance for catalysis. Two of the USPs tested naturally have an asparagine in this position, and it does not seem to affect activity. However, substituting an existing aspartate for asparagine in USP1 does result in decreased activity, suggesting that this negative charge is required in specific USPs, which might be linked to local variations in the active site. This effect can also be seen in USPs which harbor a serine in the position of the third critical residue (USP16, USP30, and USP45) (Joo et al, 2007; Gersch et al, 2017; Perez-Oliva et al, 2015; O'Dea et al, 2023). All three are catalytically competent, but the role of serine itself was only studied in USP30. When serine was substituted for asparagine and especially for aspartate, there were a major loss of catalytic activity and a dampening of its K6-linkage selectivity (Gersch et al, 2017), thus indicating that introducing a negative charge or increased steric bulk can negatively affect catalysis.

Canonically, it is thought that the fourth critical residue is involved in oxyanion hole formation (Hu et al, 2002). Interestingly, our data show that most of the tested USPs are able to perform full catalysis with the fourth critical residue in the absence of the third critical residue, and not with the third critical residue in the absence of the fourth. A dual role, with the third or fourth critical residue stabilizing catalytic histidine and oxyanion hole formation simultaneously, is unlikely, as histidine stabilization by one of the critical residues is required throughout the entire catalytic cycle. It is possible that some USPs have different requirements for oxyanion stabilization, because of minor differences in residues surrounding the catalytic cleft. USPs have an extremely conserved asparagine (USP2: N271; PDB: 2HD5; USP7: N218: PDB: 1NBF) structurally poised to stabilize oxyanion intermediates (Hu et al, 2002; Zhang et al, 2011). This asparagine alone could be sufficient for stabilizing the oxyanion intermediate in some USPs.

Direct hydrogen bonding of the fourth critical residue with histidine would not allow for catalysis, as this requires the histidine to flip, which would leave no nitrogen positioned for cysteine deprotonation. This was also shown in earlier research, where disruption of the active site by an allosteric inhibitor causes the histidine to flip (Rennie et al, 2022), resulting in inactive USP1. Instead, hydrogen bonding to the histidine would have to take place via an alternative water-mediated interaction. Through this water molecule, the fourth critical residue could act as a base, and would be able to reach catalytic histidine or even the catalytic cysteine. Although there is no structural evidence to support this, many USP structures remain to be solved and some USP structures lack the resolution to show water molecules. Interestingly, this fourth critical residue is almost always an aspartate. The high sequence and structural conservation of this fourth critical residue (aspartate) implies the importance of its negative charge. Substituting this residue to an asparagine results in an incompetent enzyme. This negative charge itself could contribute to the catalytic mechanism and promote protonation and deprotonation of the other residues involved.

Our findings are important for a fundamental understanding of the USP DUB function. This study highlights the importance of mutagenesis to characterize the catalytic triad as structural analysis alone does not explain the catalytic mechanism. Without appropriate characterization of the catalytic triad, research could be focused on the wrong residues. In addition, assumptions about the catalytic triad solely based on the canonical catalytic triad assignment in USP could affect conclusions made regarding loss-of-function mutations in genetic screens. For example, we find that some USPs retain full or most of their activity once their canonical third catalytic residue is mutated. The observed variability of mechanisms in USPs may also open up new opportunities for drug discovery. Targeting USPs is a viable strategy for the treatment of many different diseases, with promising USP inhibitors currently under development (Cadzow et al, 2020; Tsefou et al, 2021). The variety in catalytic mechanisms might allow for development of new types of inhibitors with improved specificities, selectively targeting individual USPs.

# Materials and Methods

### Plasmids, cloning, and purification

USP1/UAF1 was copurified by coexpressing USP1 (pFastbac-HTb, N-terminal His-tag, res. 21-785, G670A + G671A to prevent auto-cleavage [Huang et al, 2006]) and UAF1 (pFastbac1, N-terminal strep-tag, res. 9-677) in Sf9 cells according to Dharadhar et al (2021). USP1 had a 26-residue extension at the C-terminus (ERPLSNLE-PAVSRHAVPSLSRSTRGS), and in UAF1, the last five C-terminal residues (RQKST) have been replaced by a 21-residue extension at the C-terminus (LRSPRNSRHAVPSLSRSTRGS). A codon-optimized USP7FL (pGEX-6p-1) construct (Faesen et al, 2012) was expressed in *E. coli* and purified following the protocol described by Kim et al (2016). USP15$^{D1D2}$ (pET21a, C-terminal His-tag, res. 255-919, $\Delta$440–756, codon-optimized) was expressed in *E. coli* and purified according to Priyanka et al (2022). USP40: full-length codon-optimized USP40 was expressed in Sf9 cells and purified according to Kim R, personal communication. USP1$^{D751A}$, USP1$^{D752A}$, USP1$^{D751N}$, USP1$^{D752N}$, USP7$^{D481A}$, USP7$^{D482A}$, USP15$^{D879A}$, USP15$^{D880A}$, USP40$^{N452A}$, USP40$^{D453A}$, USP48$^{N370A}$, and USP48$^{D371A}$ in their corresponding vectors were generated with QuikChange site-directed mutagenesis, verified by sequencing, expressed and purified analogous to the WT protein. Concentrations of purified proteins were determined using the A280 measured by Nanodrop and calculated using the extinction coefficient calculated using ExPASy ProtParam (Gasteiger et al, 2005).

### Protein stability

To assess protein stability of all enzymes used in this study, a thermal stability assay was performed using nanoDSF (Prometheus NT.48; NanoTemper Technologies GmbH). In this assay, the enzymes were diluted to final concentrations of 0.5 mg/ml (USP1/UAF1, USP7) or 0.25 mg/ml (USP15, USP40, and USP48). USP1/UAF1, USP7, and USP48 were tested in 20 mM Hepes, pH 7.5, 150 mM NaCl, 5 mM DTT,

and 0.05% Tween-20 following previous publications (Faesen et al, 2011; Uckelmann et al, 2018; Dharadhar et al, 2021). USP15 (Priyanka et al, 2022) and USP40 were tested in 20 mM Hepes, pH 7.5, 100 mM NaCl, 5 mM DTT, and 0.05% Tween-20. Unfolding and aggregation of enzymes were assessed by measuring the tryptophan intrinsic fluorescence intensity over a temperature gradient from 20°C to 90°C. Using manufacturer's built-in software, the melting temperatures were determined using the ratio between fluorescence intensity values of 330 and 350 nm, and the onset of aggregation was determined using backreflection light.

### Multiple sequence alignment

The sequence of USP7's catalytic domain (residues 214-521) was used as a reference sequence to blast (Blastp, E-threshold = 10) against the UniProt database (UniProt Consortium et al, 2023). Catalytic domains as defined by UniProt of the resulting human USPs were used for the multiple sequence alignment. For USPs with multiple isoforms, the canonical isoform (isoform 1) was selected. In case of the USP17 gene family, USP17L2/DUB3 was selected (Komander et al, 2009). To properly align USP1, its inserts were removed from the catalytic domain following Dharadhar et al (2021). To properly align USP40, a shorter sequence was used (residues 250-480). Sequences were aligned using the Clustal Omega web-server (Sievers & Higgins, 2021), and alignment was visualized using Jalview software (Waterhouse et al, 2009). A sequence logo was generated based on this multiple sequence alignment of an 11–amino acid sequence surrounding the critical residues in a web-based sequence logo generator (Crooks et al, 2004).

### Structural superposition

The structural alignment was performed using USP catalytic domains for which a PDB structure is available (Table 1). Structures of USPs bound to ubiquitin were used whenever possible, to ensure a catalytically competent conformation. Structures were aligned using the ChimeraX (Goddard et al, 2018) built-in matchmaker option (Pettersen et al, 2021).

### Minimal substrate activity assays

Enzyme activities were tested on a minimal substrate consisting of ubiquitin linked to a quenched fluorophore (Ub$^{Rho}$, UbiQ). The USP40 activity was tested on Ub$^{AMC}$ (UbiQ) instead of Ub$^{Rho}$, as USP40 suffered from autoinhibition when using Ub$^{Rho}$ at higher concentrations (Kim R, personal communication). Enzyme activities were measured by the increase in fluorescence after cleavage (rhodamine: excitation at 485 nm, emission at 520 nm; AMC: excitation at 350 nm, emission at 450 nm). All reactions were performed in a 384-well plate (Corning, flat bottom, low flange) on a PHER-ASTAR plate reader (BMG Labtech) at RT. The assay buffer for USP1, USP7, and USP48 consisted of 20 mM Hepes, pH 7.5, 150 mM NaCl, 5 mM DTT, and 0.05% Tween-20. For USP15 and USP40, the assay buffer consisted of 20 mM Hepes, pH 7.5, 100 mM NaCl, 5 mM DTT, and 0.05% Tween-20.

For the kinetic analysis, we used defined enzyme concentrations of the different USPs related to intrinsic activity (USP7: 1 nM; USP1/

UAF1, USP15$^{D1D2}$, and USP40: 10 nM; and USP48: 50 nM). Each USP variant was tested against a substrate concentration series generated by twofold dilutions (USP1/UAF1 and USP40: 2,500–39.1 nM; USP7: 1,250–39.1 nM; USP15: 4,000–62.5 nM; and USP48: 3,750 to 234.4 nM). The substrate was prepared at 2x concentrations, after which 10 $\mu$l of each substrate concentration in triplicates was pipetted to the 384-well plate (Corning, flat bottom, low flange). Enzyme was injected to the plate at a 2x concentration using the PHERASTAR plate reader (BMG Labtech) built-in injector. Measurement was started after the enzyme was injected to the plate. Durations of the different experiments varied, to ensure reactions ran to completion by fully hydrolyzing Ub$^{Rho}$ or Ub$^{AMC}$ (USP1/UAF1: 1,890 s; USP7: 3,969 s; USP15: 1,764 s; and USP40 and USP48: 3,600 s).

### Michaelis–Menten analysis of USP activity

Fluorescence intensity data from the Ub$^{Rho}$ activity assays were converted to substrate concentrations using a calibration curve. For each USP, calibration curves were generated using WT enzymes, where the plateau of the completed reactions resembles the concentration of released rhodamine. The converted data of the Ub$^{Rho}$ activity assays were then analyzed using the Michaelis–Menten model in GraphPad Prism 7. First, initial velocities were determined from the linear phase of the reaction (first three datapoints). Then, to determine the kinetics, these initial velocities were plotted against the substrate concentration and then fitted using the non-linear regression Michaelis–Menten model of GraphPad Prism 7 software. To verify the results of the Michaelis–Menten analysis, we performed a global fit analysis of the data using KinTek Explorer, version 8.0 (KinTek Corporation) (Johnson, 2009) (Fig S3, Table S3).

### pH analysis

The activity of each USP was tested at pH 7.0, pH 8.0, and pH 9.0, using assay buffer with 20 mM Hepes, pH 7.0, 20 mM Hepes, pH 8.0, or 20 mM MMT, pH 9.0, replacing the 20 mM Hepes, pH 7.5. Each USP was tested against a set concentration of Ub-Rho (1 $\mu$M). Enzymes and substrates were prepared at a 2x concentration in three different buffers, and each reaction was performed in duplicates. 10 $\mu$l of enzyme in their different buffers was added to the plate. Measurement was started after 10 $\mu$l of substrate was pipetted to the plate. A full kinetic analysis was performed on USP40$^{N452A}$ and USP40$^{D453A}$, (2,500–39.06 nM) in assay buffer with 20 mM Hepes, pH 8.0, to be compared with the earlier full kinetic analysis in assay buffer containing 20 mM Hepes, pH 7.5.

### PCNA-Ub deubiquitination assays

Activities of USP1/UAF1 and USP7 variants were tested against PCNA-Ub as a more complex substrate. Mono-ubiquitinated PCNA was produced with E1 and UbcH5C$^{S22R}$ (UBE2D3) as described previously (Hibbert & Sixma, 2012) followed by purification on an S200 10/300 Increase size-exclusion chromatography column (GE Healthcare). DUB activity assays were performed at RT with 0.67 $\mu$M PCNA-Ub (trimer) in a reaction buffer composed of 20 mM Hepes, pH 7.5, 150 mM NaCl, and 2 mM DTT in a final volume of 210 $\mu$l. A 30 $\mu$l

sample was taken (T = 0-min sample) before adding an enzyme to get an accurate assessment of the ratio between PCNA-Ub and PCNA. In order to initiate the reaction, 100 nM of USP1/UAF1 or USP7 was added to remaining 180 $\mu$l reaction buffer containing 2 $\mu$M PCNA-Ub. Samples were taken at indicated time points and were added to SDS loading buffer in order to stop the reaction. Samples were loaded on a NuPAGE 4–12% Bis–Tris SDS gel and were separated by running them at 160 V for 30 min. Gels were stained using Coomassie blue and were imaged using a Gel Doc EZ imaging system (Bio-Rad Laboratories, Inc.). Using ImageLab 6.0 software (Bio-Rad Laboratories, Inc.), the volume intensities of non-ubiquitinated PCNA and ubiquitinated PCNA were measured for each time point. These volume intensities were then combined to calculate the total pool of PCNA in each lane, using which the percentage of non-ubiquitinated PCNA was determined. Because not all PCNA was ubiquitinated, all lanes were corrected for the percentage of non-ubiquitinated PCNA at $t_0$.

### Expression of USP1 in RPE1 cells

RPE1 WT and USP1 knockout cells were a kind gift from Alan d'Andrea (Lim et al, 2018). USP1 knockout cells were lentivirally transduced with a doxycycline-inducible USP1 expression vector (USP1$^{wt}$, USP1$^{C90R}$ [Morrow et al, 2018], USP1$^{D751A}$ and USP1$^{D752A}$). Cells were cultured in DMEM/F-12. Transduced cells were selected with 10 $\mu$g/ml blasticidin. Single clones were frozen for each construct. To select clones with similar USP1 levels compared with endogenous USP1 levels, single clones were incubated with 1 $\mu$g/ml doxycycline for 44 h and were lysed using RIPA buffer (1% NP-40, 1% sodium deoxycholate, 0.1% SDS, 0.15 M NaCl, 0.01 M sodium phosphate, pH 7.5, and 2 mM EDTA), containing cOmplete, EDTA-free Protease Inhibitor Cocktail (11873580001; Roche), 1 mM 2-chloro-acetamide, and 0.25 U/$\mu$l Benzonase (SC-202391; Santa Cruz Biotechnology). The total protein concentration in the lysate was determined using a BCA assay (23227; Thermo Fisher Scientific) so that equal amounts could be loaded on a gel. Samples were loaded on 4–12% Bolt gels (NW04127; Thermo Fisher Scientific), and run for 40 min at 180 V in MOPS running buffer (B0001; Thermo Fisher Scientific). Proteins were transferred to a nitrocellulose membrane (10600002; Amersham Protran 0.45 NC nitrocellulose). Membranes were stained with a USP1 antibody (14346-1-AP; Proteintech). After incubation with HRP-coupled secondary antibody, the blots were imaged using a Bio-Rad ChemiDoc XRS+. Using Bio-Rad ImageLab 5.1 software, USP1 levels were quantified by measuring the volume intensities of each USP1 band for each clone and compared with endogenous USP1 levels in RPE1 cells. Clones with comparable expression levels were selected and used for further experiments.

### Activity of USP1 mutants in RPE1 cells

The frozen single clones with comparable expression levels were taken for each construct and were incubated with or without 1 $\mu$g/ml doxycycline for 44 h. Cells were lysed following the Materials and Methods section described above. To load equal amounts on a gel, the total protein concentration in the lysate was determined using the previously mentioned BCA assay. Samples were loaded, and gels were run following the same procedure as above. Proteins

were transferred to a nitrocellulose membrane (10600002; Amersham Protran 0.45 NC nitrocellulose). Membranes were stained with the following antibodies: Ubiquityl-PCNA (Lys164) (D5C7P) (13439; Cell Signaling Technology), PCNA (PC10) (sc-56; Santa Cruz Biotechnology), and USP1 (14346-1-AP; Proteintech). After incubation with HRP-coupled secondary antibodies, the blots were imaged using a Bio-Rad ChemiDoc XRS+. Using Bio-Rad ImageLab 5.1 software, we quantified PCNA-Ub levels by measuring the volume intensities of each PCNA-Ub band for each clone before and after doxycycline induction, and ratios between these PCNA-Ub volume intensities were calculated. The distributions of these ratio values for different constructs were compared using non-parametric one-way ANOVA (Kruskal–Wallis test) corrected for multiple comparison by controlling the false discovery rate (two-stage linear step-up procedure of Benjamini, Krieger, and Yekutieli) in GraphPad Prism V9.5.

### Ubiquitin-propargyl assays

Each USP variant (wt and mutants) was incubated at a single concentration (2 $\mu$M) with 16 $\mu$M of ubiquitin-propargyl (Ub$^{PA}$, UbiQ) in conjugation buffer (20 mM Tris, pH 8.0, 150 mM NaCl, and 2 mM TCEP) at RT. Enzymes and substrates (Ub$^{PA}$) were prepared at a 2x concentration in conjugation buffer and were combined to initiate the reaction. As a reference, WT of each enzyme at 2x concentration was incubated with conjugation buffer instead of Ub$^{PA}$. After 5 min, samples were taken and added to SDS loading buffer to terminate the reaction. For USP1 and USP48, samples were taken after 60 min as well. Samples were run on a 4–12% gradient gel.

### Stopped-flow analysis of initial rates

To gain insights into the initial rates, a single concentration of enzyme (USP1/UAF1, USP7, USP15$^{D1D2}$, and USP40: 10 nM; USP48: 50 nM) was tested against a 10-fold higher concentration of substrate (Ub$^{Rho}$: 100 nM or 500 nM in case of USP48). Here, Ub$^{Rho}$ was used for USP40 instead of Ub$^{AMC}$ because Ub$^{Rho}$ does not cause substrate inhibition at low concentrations (Kim R, personal communication) and because Ub$^{AMC}$ is more prone to photobleaching. Enzymes and substrates were prepared at 2x concentrations and were prepared in their corresponding buffers mentioned in the minimal substrate activity assays. Equal amounts of a stock solution of ubiquitin–substrate and of USP proteins were mixed in an SF-61DX2 stopped-flow fluorimeter system (TgK Scientific Ltd) with R10699 photomultipliers (Hamamatsu Photonics K.K.). Measurements were carried out for 100 s at RT, with each measurement repeated three times.

# Supplementary Information

# Acknowledgements

The authors would like to thank Farid El Oualid, other members of the department, and Herbert Waldmann and David Barford for useful discussions. Research at the Netherlands Cancer Institute was supported by institutional grants from the Dutch Cancer Society and from the Dutch Ministry of Health, Welfare and Sport. Funding was provided by NWO (OCENW.KLEIN.131, LIFT 731.017.415, NWO-Echo 711.017.008, to TK Sixma), Oncode Institute (to TK Sixma), and the German Research Foundation (DFG, GE 3110/1-1, to M Gersch).

## Author Contributions

N Keijzer: conceptualization, data curation, formal analysis, investigation, visualization, methodology, and writing—original draft, review, and editing.
A Priyanka: conceptualization, investigation, methodology, and writing—review and editing.
Y Stijf-Bultsma: investigation, methodology, and writing—review and editing.
A Fish: formal analysis, methodology, and writing—review and editing.
M Gersch: validation and writing—review and editing.
TK Sixma: conceptualization, supervision, funding acquisition, methodology, and writing—review and editing.

## Conflict of Interest Statement

The authors declare that they have no conflict of interest.

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
