## [Reviewer comments · Life Science Alliance]

Life Science Alliance

Variety in the USP deubiquitinase catalytic mechanism

Niels Keijzer, Anu Priyanka, Yvette Stijf-Bultsma, Alex Fish, Malte Gersch, and Titia Sixma

DOI: <https://doi.org/10.26508/lsa.202302533>

Corresponding author(s): Titia Sixma, Netherlands Cancer Institute-Antoni van Leeuwenhoek Hospital

Review Timeline:

Submission Date:	2023-12-14
Editorial Decision:	2024-01-08
Revision Received:	2024-01-23
Accepted:	2024-01-23

Transaction Report:

Please note that the manuscript was reviewed at *Review Commons* and these reports were taken into account in the decision-making process at *Life Science Alliance*.

Review
COMMONS

Reviews

Review #1

****Summary:****

The authors study the functional role of two adjacent active site residues as candidates for polarising the catalytic histidine in the "Asn/Asp" box from five phylogenetically unrelated ubiquitin specific proteases (USP1, USP7, USP15, USP40 and USP48). One of these residues is more variable across USPs (Asn, Asp, Ser), whereas the second one is absolutely conserved (Asp). To this end they use alanine mutants in kinetic experiments and test their ability to crosslink to ubiquitin propargyl as a proxy for testing the nucleophilicity of the catalytic cysteine. They then further evaluate the activity of the USP1 mutants in processing PCNA-Ub in RPE1 cells.

They find that the role of these two residues differs between the different USPs studied, which is in line with previous work that has shown that in USP7, the amongst USPs less conserved residue takes on the major role of polarising the histidine, whereas in the more distantly related USP2, the absolutely conserved Asp is more important (Zhang W, et al. Contribution of active site residues to substrate hydrolysis by USP2: insights into catalysis by ubiquitin specific proteases. *Biochemistry*. 2011 50(21):4775-85. doi: 10.1021/bi101958h). This study expands on these findings to evaluate the role of these residues in four other USPs.

****Major comments:****

1. The authors compare highly diverse USPs; USP1 requires UAF1 for full activity and the complex is used in the study, USP7 requires a C-terminal tail peptide for full activity, USP40 and USP48 belong to the CHN class, whereas USP7, USP15 and USP1 belong to the CHD class of USPs. The rationale for selecting this diverse set of USPs is therefore not clear and makes direct comparisons of the findings more difficult. It is certainly interesting that the previously published differences between USP2 and USP7 with respect to these residues are also found in four other divergent USPs, but for this reason it isn't as "surprising" as the title suggests. The title, omission of background knowledge on USP2 in the abstract and presentation of the findings in a graph that makes direct comparisons (Figure 5) are therefore a bit misleading, which needs addressing.
2. The study relies on single alanine mutations, which will inevitably change the hydrogen bonding patterns and the local environment which could impact the conclusion. The authors should verify in kinetic assays at least for USP1, which is the main focus, that Asp to Asn mutants still display the same effects.
3. While neither mutant unfolds below 40 degrees, there are clear differences in thermal stability between some of the proteins used in the study (Supp. Fig. 1B). A full table of measured Tms by NanoDSF for all Wt and mutant proteins should be provided so that the reader can evaluate how the results may be impacted by local effects that impact the thermal stability. It is noticeable that USP40 and USP15 mutants in particular display large differences in thermal stability, which could directly affect the results. The authors should clearly discuss these limitations of the study.

****Minor comments:****

1. For USP48 and USP40 no published structures are available at present, so it isn't clear whether there are any differences in orientation of the studied residues. An unpublished USP40 structure is referred to but not shown. The general conclusion that structures do not reveal any differences in these residues may therefore not be valid for all the studied USPs. Please revise.
2. The introduction of the new terms "critical residue 1 and 2" are confusing and partially disproved by the study itself (replace with e.g. less conserved versus absolutely conserved 3rd triad residue or similar), please revise.
3. p. 3/4: please add pH information to buffers used in the stability studies. "Previous publication" and "manuscript in preparation" are contradictions.
4. p. 4. Assay buffer for USP1, USP7 and USP48 pH information is missing
5. p. 6: last heading: typo is dispensable
6. p. 8: please explain choice of USP1 C90R mutation
7. Explain choice of pH range 7-9 studied with regards to anticipated pKas
8. Importance of mutagenesis for studying enzymatic mechanisms is clear but limitations also need to be discussed; introduction of local changes etc.. this should be added to the discussion
9. Table 1: linear not linear
10. Table 2: add information for mutant names (exact residue numbers) these data correspond to to improve clarity
11. Fig. 1D which structure is shown?
12. Fig. 4 bands for USP1/UAF1 D752A and USP15 WT/mutants very faint so difficult to see whether there is crosslinking or not, please comment

13. Fig.5: please see above for comment about graph and remove or revise.

14. Suppl. Table 2: global fit analysis not appropriate for when a poor fit was obtained or where the mutants were barely active (Figs S2, S3). These constants should be removed from the table or more information on the fitting provided. There seems to be some correlation between barely active mutants and the thermal stability, please comment.

15. Suppl. Fig. 1B: See above.

****Referees cross-commenting****

reviewers' comments are balanced

The study builds on previous work on USP7 and USP2 and while not a conceptual advance, adds to our understanding and knowledge of USP mechanisms. The in cellulo work of probing critical residues in USP1 for processing PCNA-Ub adds a new dimension.

However, the limitations of some of the experimental design, stability of mutants and choice of USPs (as outlined above) somewhat hamper the direct comparisons the study makes and previous work needs to be adequately represented (USP2). The work will be of interest to basic researchers and medicinal chemists in particular.

Review #2

Dr. Sixma is a leading expert in DUB enzymology, especially the enzymology of USP family members. This manuscript is a welcome addition to the field and her body of work to date. Exploring the possibility of redundant or entirely new catalytic residues in USPs is indeed an important venture for differentiating these highly homologous enzymes. The paper is well-written, and the experiments are simplistic and understandable. However, as a whole, the work is not ground-breaking, and the mechanistic explanation of the experimental observation lacks substantiating evidence. The manuscript should be recommended for publication in an appropriate journal after some revision.

****Major comments:****

- A major concern of the article is about the mechanistic explanation of the role of the second critical residue Asp. The authors proposed two different possible mechanisms, including 1. the residue is flexible to position itself to replace the role of the canonical general base "first" critical residue; 2. Cys/His forms a dyad as seen in other cysteine proteases, and the "second critical residue" Asp participates in the oxyanion hole to stabilize the activated substrate. However, as the authors argue in their discussion, both mechanisms are speculative and have major issues: mechanism #1 requires the catalytic His to flip, and the conformation of the His and "second" critical residue is not optimal for them to form a hydrogen bond directly. The author suggested it may be mediated by a water molecule. However, no such structure has been reported. Mechanism #2 also has the trouble of lacking experimental evidence, and since the tetrahedral oxyanion intermediate is negatively charged, the same negatively charged Asp would be unfavourable. Without mechanistic evidence, the observation of the second (more) critical residue Asp is a very interesting one but beyond that, most of the discussions are speculative. The activity-based labelling experiment using Ub-PA, and the cellular experiments using the mutants only confirmed the observation but can not approve any of these mechanisms.

- The possibility of substrate trapping in some mutants is of interest. Paragraph 5 of the discussion even mentions this. I think this should be investigated by single-turnover assay techniques.

****Minor general concern:****

- The naming of the Asp/Asn/Ser in the canonical triad is a bit confusing. It is called "the third catalytic residue" and then the "first critical residue" (Intro, last paragraph). This is confusing because, in the catalytic triad, Cys/His are also critical residues. Given the importance of the fourth Asp residue, maybe the authors should come up with a different naming system. One suggestion could be calling the Asp/Asn/Ser the ****general base residue**** (in the canonical triad terms, Cys is the nucleophile, His is the general acid-base residue, Asp is the general base residue), and the 4th Asp as the "alternative general base residue"?

- The augment at the end of the discussion that this alternative Asp residue could lead to new inhibitors for this difficult class of cysteine proteases is a stretch. The majority, if not all, structurally defined inhibitors of USPs (USP7, USP1, USP14) are allosteric inhibitors that do not target the catalytic triad directly. I doubt the discovery of Asp will change that. The most variability of activity regulation of USPs comes from auxiliary domains of the FL USPs, or cofactor proteins, as the authors' lab has previously demonstrated for many of the USPs, including USP7, USP4, USP1, etc., and there lie more opportunities for new inhibitor discovery.

- Similarly, it is a fancy term to cite of DUBTACs, but I don't see much relevance of this alternative residue applied to DUBTACs. The authors could explore the idea a bit if they decide to cite this.

Minor comments and grammar: editing is difficult without the inclusion of line numbers. I have attempted to address errors the best I can, considering this.

- Synopsis: "..., the majority of USPs **does** not..." should be "**do**"
- Synopsis: "..., either critical **residues** can..." should be "**residue**"
- Intro: "Subsequently a tetrahedral..." should have a comma after subsequently
- Intro: 2nd paragraph, line 6, be more specific to be "peptide bond."
- Intro: in the 3rd paragraph, the residue numbers of the catalytic residues should be stated.
- Intro: the first line of paragraph 4. The statement is confusing and should be made clearer by simply stating, "The third catalytic residue in USPs is either Asp, Asn, or Ser."
- Intro: second last paragraph, be a bit more specific on what "resembles USP15 and USP7" could be "... USP8, another USP whose catalytic triad resembles those of USP15 and USP7" because the domain structure of these FL USPs is very different, only the triad is similar.
- Intro: the last paragraph mentions the loss of function USP15 mutation behaves like wild type and USP1. The term "loss of function" is misleading. If mutation to the canonical 3rd catalytic residue has no effect on activity, then it is not a loss of function mutant. Please specify the alanine mutation.
- Intro: last paragraph, "Michaelis Menten," should have a hyphen in between.
- Methods: please add a space between values and units; this comes up multiple times throughout the manuscript
- Methods: all taxonomic names should be italicized, i.e., *E. coli*
- Methods: protein stability section, "**build-in**" should be "**built-in**" (build-in is repeated elsewhere and needs to be fixed)
- Methods: structure superposition section, "... bound to ubiquitin were **use** whenever..." should be "...bound to ubiquitin were **used** whenever..."
- Methods: pH analysis section, "duplo" should be duplicate
- Methods: Expression of USP1 in RPE1 cells section, please briefly state how you determined the expression level of USP1 in transduced RPE1 USP1KO cells when selecting clones with comparable levels to RPE1 wt cells
- Methods: tCoffee webserver should be "T-Coffee"
- Methods: MSA. Can the authors provide more details on when doing BLAST, what were the criteria of selecting sequences from the result?
- Methods: please provide the details for determining the concentration of the enzymes used.
- Methods: Please provide the manufacturers of the Pherastar plate reader and the 384-well plate (please correct from "384 well-plate").
- Results: In paragraph 1, "lies a **much better** conserved..." you should use "more highly."
- Results: paragraph 1, "USP50 does not harbor either of" should be "USP50 harbors neither of"
- Supp Fig 2: USP39 does not have glutamate in position of the first critical residue, it is glutamine (Q)
- Results: second subsection title "**The first critical residue is dispenUSP1...**" needs to be fixed
- Results: pg. 8 last line "to crosslink", the word crosslink is not proper for the reaction between Ub-PA with USPs. It usually refers to a reactive linker that links two molecules. Words like "conjugate", "conjugation," or "covalent react with", and "activity-based labelling" are probably better choices depending on the context.
- Figure 1: figure legend describing B, C, and D are mixed up.
- Results: In paragraph 9, the statement that your data on 5 USPs is representative of most of the 57 members in that the third catalytic is dispensable is not a sound statement for the small sample size. I think more emphasis on the diversity of USP1, USP7, USP15, USP40, and USP48 needs to be stated to help bolster such a claim. The statement to follow, which mentions sequence analysis alone is not able to predict the catalytic residue, is also somewhat contradictory to the opening statement and insinuates that all active USPs should be tested, while you only examined 5.
- Figure 4: legend title, the critical residues are not responsible for **_performing_** nucleophilic attack per se; that is the job of Cys. The title of the figure should be altered to clear this up.
- Discussion: paragraph 3, since the Hu 2002 USP7 mechanism is not valid for other USPs tested, the "consensus USP catalytic mechanism" should be referred to as the "canonical."
- Discussion: paragraph 4, "USP7, USP15 and USP40 all **three** have misaligned..." should be "USP7, USP15 and USP40 all have misaligned..."
- Discussion: paragraph 8, "negative charge itself could **contributes**..." should be "negative charge itself could **contribute**..."
- Discussion: pg. 10, 3rd paragraph. Is the first sentence a statement of fact or a hypothesis? The writing is not clear to differentiate the two possibilities.
- Discussion: pg. 10, 3rd paragraph, line 3, which "critical residue" does it refer to, the general base residue or the alternative residue?
- Discussion: pg. 10, second last paragraph. Can the statement that "inaccurate assumptions about the catalytic triad ... be substantiated with an example?"

- Table 1, "ubiquitin variant" is mostly often used in the literature to refer to the ubiquitin mutants generated by phage display pioneered by the Sidhu lab or designed mutants. "ubiquitin and homolog derivatives" is a better term for "ubiquitin variant" in this article.
 - Table 1, the USP21 line "Lineair" is a typo, it should be "linear."
 - References: citations for Cadzow, 2020. and Tsefou, 2021 do not appear in the bibliography.
 - Add a hyphen to "Ubiquitin-specific proteases."
- *General assessment:*

Based on the studies of prototypical ubiquitin-specific protease USP7, the field generally accepts that USPs are a class of cysteine proteases that contain a catalytic triad with a cysteine, a histidine and a general base residue (asparagine, aspartate, or serine). This manuscript described the importance of an alternative, highly conserved aspartate that plays a critical role in catalysis using an enzyme kinetics study on five out of 57 USPs. The work is a very interesting observation that could change the perception in the field. However, the atomic details of how this fourth, or alternative residue, plays its role in catalysis are not clear without the structure evidence of an intermediate/transition state-bound complex.

Advance:

The study provided the first systematic enzymology study of the role of a fourth conserved residue critical for the catalysis of USPs. It is a conceptual advance and a first step to elucidate possibly a new catalytic mechanism of USPs.

Audience:

The manuscript will be of interest to biochemists in the field of ubiquitination and drug discovery.

Reviewers' expertise

The reviewers are structural biologists with expertise in the structure, function and enzymology of ubiquitin enzymes in general, with practical experience in drug discovery targeting the DUB and kinase families.

Less than 1 month

Yes

Revision 0, Review #3 - Rittinger, Katrin (katrin.rittinger@crick.ac.uk)

The article by Keijzer and colleagues describes an interesting study comparing the active site of multiple USPs (the largest subfamily of deubiquitinases) and elucidating the importance of specific residues lining the active site for catalysis. The authors carried out a careful analysis of the kinetic properties of 5 representative USPs and mutants thereof revealing a remarkable variety in their function that highlights that the majority of USPs studied do not require the canonical third residue of the catalytic triad of USPs for activity but instead rely on a highly conserved second critical residue. Furthermore, the authors apply complementary experimental approaches (mutagenesis, pH dependence of activity, crosslinking with Ub-PA) to allow distinguishing between residues important for the nucleophilic attack versus oxyanion hole stabilisation.

This is a well-written, thorough enzymatic study of high technical quality. The experiments are described in sufficient detail to allow others to reproduce the experimental set up. The data presented fully support the claims of the paper and no additional experiments are required to further support the conclusions.

It is great to see that the authors have carried out thermal stability assays on all WT and mutant proteins under investigation to ensure that any effects observed are not due to protein misfolding.

Minor comments:

- There are a few typos in the manuscript the authors should correct.
- The panels/paper legends to Figure 1B/C/D are mixed up. Please correct.
- It would be helpful to use different colours in the alignment shown in Supplementary Figure1 to indicate the position of the first and second critical residue.
- I wonder if the authors could comment on how representative the 5 USPs characterised in this work are of the entire family.

Deubiquitinating enzymes (DUBs) play essential roles in many cellular processes and their activity is associated with a variety of diseases. There is a lot of interest in targeting DUBs for therapeutic purposes and a number of small molecule inhibitors are undergoing clinical studies.

While the structure and mechanism of multiple DUBs have been studied over the years, many open questions about their detailed catalytic mechanism remain and the importance of specific residues might often have been inferred based on sequence conservation alone without accompanying experimental support.

This work makes an important contribution to the field by systematically examining 5 members of the USP family and defining the precise role of the first and second critical residue for the catalytic cycle. This work will be of interest to those studying the mechanism of DUBs in general and those trying to target specific DUBs with small molecules. In addition, this study will also be interesting more generally for those studying enzyme kinetics as it highlights the importance of experimental validation of a catalytic mechanism that has been predicted based on sequence conservation or structural studies.

1. General Statements

We thank the reviewers for their constructive and detailed reviews. We have been able to resolve all issues raised by the reviewers with additional experiments and changes in the text:

- In response to two of the reviewers we've changed the nomenclature of the residues. As we would like to avoid assigning roles in the naming, we now use 'critical residue 3' and 'critical residue 4', with Cys and His forming critical residue number 1 and 2 respectively.
- We analyzed the role of the negative charge in the fourth critical residue of USP1, by mutating this Asp to Asn to assess the importance of a charged residue in these positions (Supplementary figure 2), resulting in complete loss of activity just like the alanine mutant. We also tested the effect of mutating the third critical residue to Asn in USP1, which causes a minor decrease in activity. This highlights the importance of the highly conserved aspartate (fourth critical residue), and shows that precise residue found in the position is important for catalysis. Additionally, these mutants address potential effects of the 'holes' left by the original Ala mutations.
- Importantly, we were able to perform single-turnover assays to expand on our analysis of the precise roles of the critical residues and give more fundamental insight in the defects of the mutants. These assays further elaborate on the variability observed between these USPs. In USP15, these experiments explain the defect in catalysis for the third critical residue mutant and provides insight how a successful nucleophilic attack is combined with defective catalysis (updated Figure 4), which is not observed in the other USPs we tested. In these other USPs, the single turnover experiments reveal that the nucleophilic attack performed by the third and fourth critical residue mutant of USP7 and USP40 happens with low efficiency, even lower efficiency for USP48 and that this ability is lost entirely in USP1.
- We included a number of important textual changes to better explain the choices and variation in USPs tested, highlight prior USP2 data and the implications for drug discovery.
- We updated Ub-PA conjugation assays (updated Figure 4) for better contrast, and repeated the Ub-PA assay for USP1 and USP48 with longer incubation (Supplementary Figure 6).

More details are given in the point-by-point response below. All in all, we are convinced that this much improved manuscript is now ready for publication and hope that all reviewers will agree.

Reviewer #1 (Evidence, reproducibility and clarity (Required)):

Summary:

The authors study the functional role of two adjacent active site residues as candidates for polarising the catalytic histidine in the "Asn/Asp" box from five phylogenetically unrelated ubiquitin specific proteases (USP1, USP7, USP15, USP40 and USP48). One of these residues is more variable across USPs (Asn, Asp, Ser), whereas the second one is absolutely conserved (Asp). To this end they use alanine mutants in kinetic experiments and test their ability to crosslink to ubiquitin propargyl as a proxy for testing the nucleophilicity of the catalytic cysteine. They then further evaluate the activity of the USP1 mutants in processing PCNA-Ub in RPE1 cells.

They find that the role of these two residues differs between the different USPs studied, which is in line with previous work that has shown that in USP7, the amongst USPs less conserved residue takes on the major role of polarising the

histidine, whereas in the more distantly related USP2, the absolutely conserved Asp is more important (Zhang W, et al. Contribution of active site residues to substrate hydrolysis by USP2: insights into catalysis by ubiquitin specific proteases. *Biochemistry*. 2011 50(21):4775-85. doi: 10.1021/bi101958h). This study expands on these findings to evaluate the role of these residues in four other USPs.

Major comments:

1. The authors compare highly diverse USPs; USP1 requires UAF1 for full activity and the complex is used in the study, USP7 requires a C-terminal tail peptide for full activity, USP40 and USP48 belong to the CHN class, whereas USP7, USP15 and USP1 belong to the CHD class of USPs. The rationale for selecting this diverse set of USPs is therefore not clear and makes direct comparisons of the findings more difficult. It is certainly interesting that the previously published differences between USP2 and USP7 with respect to these residues are also found in four other divergent USPs, but for this reason it isn't as "surprising" as the title suggests. The title, omission of background knowledge on USP2 in the abstract and presentation of the findings in a graph that makes direct comparisons (Figure 5) are therefore a bit misleading, which needs addressing.

- We apologize that it seemed as if we had overlooked USP2, for which both critical residues are important, and we agree that our abstract previously focused too much on the perception of the field and its focus on USP7. We have changed the abstract and introduction to highlight the USP2 data for a more balanced perspective.
- The reviewer is correct that the set of USPs is diverse, but we see this as a strength, given that this is the first manuscript in which these residues are analyzed in a comparative side-by-side manner for multiple DUBs. We find that our results are not directly related to the CHN/CHD diversity (i.e. changes in the third catalytic residue), nor apparently to activation by a C-terminal tail (as both USP7 and USP40 have this mechanism). Since these are structurally conserved enzymes with a common fold, we do find the comparison is informative. Furthermore, we felt that it was important to clearly signal the variation in different steps of the mechanism, something which appears to largely remain unnoticed by the field. Figure 5 is helpful in understanding that these changes have multiple dimensions. We agree that it is important to signal the diversity as possible source for these differences and we have added the following sentences to paragraph 3 of the results: "These USPs vary in domain architecture and allosteric regulation, and therefore represent different aspects of the USP family. USP1, USP7 and USP15 both harbor two aspartates as third and fourth critical residue and USP40 and USP48 harbor an asparagine and aspartate as third and fourth critical residue respectively, allowing us to examine the importance of a negative charge in position of the third critical residue."
- We used the word surprising in the title to indicate the variability we observed in the two dimensions of the mechanism, as indicated in Fig. 5.

2. The study relies on single alanine mutations, which will inevitably change the hydrogen bonding patterns and the local environment which could impact the conclusion. The authors should verify in kinetic assays at least for USP1, which is the main focus, that Asp to Asn mutants still display the same effects.

- We are thankful for this suggestion. We have made these additional USP1 mutants through insect cell expression and tested these in different assays. As expected, both Asn mutants follow the alanine mutations. The results are reported in Supplementary fig 2BC.

3. While neither mutant unfolds below 40 degrees, there are clear differences in thermal stability between some of the proteins used in the study (Supp. Fig. 1B). A full table of measured T_{ms} by NanoDSF for all Wt and mutant proteins should be provided so that the reader can evaluate how the results may be impacted by local effects that impact the thermal stability. It is noticeable that USP40 and USP15 mutants in particular display large differences in thermal stability, which could directly affect the results. The authors should clearly discuss these limitations of the study.

- We have added supplemental table S2 to report the melting temperatures. The effect observed for USP15 is addressed in the results: "While both mutants of USP15 have a decreased thermal stability compared to USP15^{wt}, these variants retain stability until 50 °C, indicating that they are still well-folded and suitable for

kinetic assays at room temperature.” For USP40, it is not the actual measured T_m that deviates a lot, but the measured 350/330 ratio, which is addressed in the legend of supplementary figure 1B “Ratios measured (350 nm/330 nm) varied between some of the mutants (Eg. USP40wt), but this did not affect the measured inflection points (Supplementary table 1)”.

Minor comments:

1. For USP48 and USP40 no published structures are available at present, so it isn't clear whether there are any differences in orientation of the studied residues. An unpublished USP40 structure is referred to but not shown. The general conclusion that structures do not reveal any differences in these residues may therefore not be valid for all the studied USPs. Please revise.

- We apologize if this was not clear. We did however not refer to a USP40 structure, but a USP40 manuscript in preparation that studies biochemistry USP40 activation through activation by its C-terminal tail.
- The existing structures do not show observable differences in the active site residues, nor in the immediate surrounding, and therefore do not give insight which residue is critical for catalysis. We now mention this more explicitly. “It was previously shown that there are no structural differences in the positioning of the catalytic triad and the fourth critical residue between USP2 and USP7, despite their third and fourth critical residues behaving differently (Zhang et al., 2011). We superimposed the currently available crystal structures of USP catalytic domains (Table 1, Figure 1E) and also found only minor differences in the positioning of these two adjacent residues.”
- As the AlphaFold predictions for USP40 and USP48 closely resemble the known structures in Figure 1E, we have added this information as follows : “While the structures of USP40 and USP48 have not been solved, they contain the conserved USP catalytic domain and AlphaFold predictions for USP40 (Uniprot: Q9NVE5) and USP48 (Uniprot: Q86UV5) do not suggest major changes in their catalytic domains.”

2. The introduction of the new terms "critical residue 1 and 2" are confusing and partially disproved by the study itself (replace with e.g. less conserved versus absolutely conserved 3rd triad residue or similar), please revise.

- Thank you, this issue is also mentioned by reviewer 2. We aimed at a solution that would not make inferences on mechanisms. We settled on "critical residue 3" and "critical residue 4", with the active site Cys and His being the first two.

3. p. 3/4: please add pH information to buffers used in the stability studies. "Previous publication" and "manuscript in preparation" are contradictions.

- pH information has been added.
- Thank you for the comments, we've adjusted the text.

4. p. 4. Assay buffer for USP1, USP7 and USP48 pH information is missing

- We have corrected the omission.

5. p. 6: last heading: typo is dispensable

- Typo was corrected.

6. p. 8: please explain choice of USP1 C90R mutation

- Other mutations tend to increase affinity for free ubiquitin, and in cells this can change ubiquitin homeostasis. The Cys to Arg mutation was shown to avoid this problem in some DUBs. (Morrow et al, Embo Rep 2018 Oct;19(10):e45680. doi: 10.15252/embr.201745680). We have added the reference in both the methods and results sections.

7. Explain choice of pH range 7-9 studied with regards to anticipated pKas

- We primarily aimed to look at the catalytic cysteine, which needs to be deprotonated in order to allow for catalysis. The sentence on pKas has been removed to avoid confusion. Since the catalytic cysteine in USPs

typically has a high pKa, we decided to look at an increased pH to favor partial Cys deprotonation. To that, we have added a reference on USP7, in which it was previously shown USP7 is activated by a higher pH, which holds true for both full-length and its catalytic domain (Faesen et al., (2011). *Molecular Cell*, 44(1), 147–159. <https://doi.org/10.1016/j.molcel.2011.06.034>).

8. Importance of mutagenesis for studying enzymatic mechanisms is clear but limitations also need to be discussed; introduction of local changes etc.. this should be added to the discussion

- We have extended the discussion of limitations as requested. Importantly, the new USP1 asparagine mutants relieve some of the limitations of using alanine substitutions, which we also addressed in this section of the discussion: “While alanine mutations leave open an empty space, or take away the negative charge whenever an aspartate is mutated, mutating both critical residues to asparagine in USP1 did not alleviate the decrease in catalytic competence. Additionally, all single critical residue mutants remained stable and some mutants retaining most of their catalytic competence suggests that these enzymes still function properly.”

9. Table 1: linear not linear

- Thank you. We have made the change.

10. Table 2: add information for mutant names (exact residue numbers) these data correspond to to improve clarity

- Thank you. We have made the change.

11. Fig. 1D which structure is shown?

- USP7 (1NBF), we have adjusted the legend.

12. Fig. 4 bands for USP1/UAF1 D752A and USP15 WT/mutants very faint so difficult to see whether there is crosslinking or not, please comment

- We performed the experiment again and made new figures with better contrast.

13. Fig.5: please see above for comment about graph and remove or revise.

- We have adjusted the legend to make the diversity more clear: “These five USPs share the conserved USP catalytic domain but vary considerably in domain architecture and allosteric regulation, and therefore represent a part of the diversity found in the USP family.”

14. Suppl. Table 2: global fit analysis not appropriate for when a poor fit was obtained or where the mutants were barely active (Figs S2, S3). These constants should be removed from the table or more information on the fitting provided.

There seems to be some correlation between barely active mutants and the thermal stability, please comment.

- We prefer to do the global fit analysis, as it enables us to share rate constants and get meaningful comparisons. All USP variants were fit simultaneously using the global fit approach where k_1 and k_3 rate constants were fixed, k_{-1} and k_{-3} were shared for all the data sets of the same USP and only k_2 was fitted for each data set separately. The quality of the global fit correlates with standard errors of k_{-1} and k_{-3} rate constants. So, the model we use fits reasonably well with all the data sets all together. Even though a few fitted curves are not aligned well with some of the data for mutants with low activity the value of k_2 is still important to report since it gives an approximation of magnitude for the catalytic activity and high standard error reflects the quality of the fit for those specific data sets. In addition, k_{cat}/K_m values for all the proteins, including low activity mutants, calculated from global fit approach correlate well with the values calculated from Michaelis-Menten analysis. We clarified this in the legend of supplementary figure 3: “Our kinetic model fits the data well. No fit could be obtained for USP15^{D880A} since no activity was detected. We got relatively poorer fits to USPs with low activity, USP1^{D752A}, USP7^{D481A}). Still, for these low activity USPs the reported K_{cat}/K_m gives an approximation of the magnitude for the catalytic activity and the poorer fit is reflected by their relatively higher standard errors reported in supplementary table 3.”

15. Suppl. Fig. 1B: See above.

- See comment on 3.

****Referees cross-commenting****

reviewers' comments are balanced

Reviewer #1 (Significance (Required)):

The study builds on previous work on USP7 and USP2 and while not a conceptual advance, adds to our understanding and knowledge of USP mechanisms. The in cellulo work of probing critical residues in USP1 for processing PCNA-Ub adds a new dimension.

However, the limitations of some of the experimental design, stability of mutants and choice of USPs (as outlined above) somewhat hamper the direct comparisons the study makes and previous work needs to be adequately represented (USP2). The work will be of interest to basic researchers and medicinal chemists in particular.

- We very much appreciate the enthusiasm of the reviewer for our cellular validation.

Reviewer #2 (Evidence, reproducibility and clarity (Required)):

Dr. Sixma is a leading expert in DUB enzymology, especially the enzymology of USP family members. This manuscript is a welcome addition to the field and her body of work to date. Exploring the possibility of redundant or entirely new catalytic residues in USPs is indeed an important venture for differentiating these highly homologous enzymes. The paper is well-written, and the experiments are simplistic and understandable. However, as a whole, the work is not ground-breaking, and the mechanistic explanation of the experimental observation lacks substantiating evidence. The manuscript should be recommended for publication in an appropriate journal after some revision.

Major comments:

- A major concern of the article is about the mechanistic explanation of the role of the second critical residue Asp. The authors proposed two different possible mechanisms, including 1. the residue is flexible to position itself to replace the role of the canonical general base "first" critical residue; 2. Cys/His forms a dyad as seen in other cysteine proteases, and the "second critical residue" Asp participates in the oxyanion hole to stabilize the activated substrate. However, as the authors argue in their discussion, both mechanisms are speculative and have major issues: mechanism #1 requires the catalytic His to flip, and the conformation of the His and "second" critical residue is not optimal for them to form a hydrogen bond directly. The author suggested it may be mediated by a water molecule. However, no such structure has been reported. Mechanism #2 also has the trouble of lacking experimental evidence, and since the tetrahedral oxyanion intermediate is negatively charged, the same negatively charged Asp would be unfavourable. Without mechanistic evidence, the observation of the second (more) critical residue Asp is a very interesting one but beyond that, most of the discussions are speculative. The activity-based labelling experiment using Ub-PA, and the cellular experiments using the mutants only confirmed the observation but can not approve any of these mechanisms.

- Indeed, we do not come with a full mechanistic explanation which explains catalysis in all USPs. Instead, we show that individual USPs have greatly different dependence on their catalytic residue, and thus display important mechanistic distinctions, both for nucleophilic attack and for completion of the reaction. The new Asn mutations do show that negative charge in the 4th critical residue is critical for USP1 function, while the new stopped-flow

analysis reveals that USP15 is trapped after the first turnover when the 4th critical residue is lost, and that this is not the case for the other USPs tested.

- The possibility of substrate trapping in some mutants is of interest. Paragraph 5 of the discussion even mentions this. I think this should be investigated by single-turnover assay techniques.

- We are very thankful for this great suggestion. We performed fast kinetics assays (stopped flow) for all USP wildtype and alanine variants. Together with the Ub-PA labelling experiments these assays shed new light on the ability of these USPs to perform a nucleophilic attack. In terms of substrate trapping, it does indeed turn out that USP15 is inactivated after the first turnover (Figure 4B).

Minor general concern:

- The naming of the Asp/Asn/Ser in the canonical triad is a bit confusing. It is called "the third catalytic residue" and then the "first critical residue" (Intro, last paragraph). This is confusing because, in the catalytic triad, Cys/His are also critical residues. Given the importance of the fourth Asp residue, maybe the authors should come up with a different naming system. One suggestion could be calling the Asp/Asn/Ser the ****general base residue**** (in the canonical triad terms, Cys is the nucleophile, His is the general acid-base residue, Asp is the general base residue), and the 4th Asp as the "alternative general base residue"?

- Reviewer 1 also did not like the naming. To address the issue we have settled on: "critical residue 3" and "critical residue 4", with the active site Cys and His being the first two. This avoids assigning mechanistic roles to particular residues, but still stresses their importance.

- The augment at the end of the discussion that this alternative Asp residue could lead to new inhibitors for this difficult class of cysteine proteases is a stretch. The majority, if not all, structurally defined inhibitors of USPs (USP7, USP1, USP14) are allosteric inhibitors that do not target the catalytic triad directly. I doubt the discovery of Asp will change that. The most variability of activity regulation of USPs comes from auxiliary domains of the FL USPs, or cofactor proteins, as the authors' lab has previously demonstrated for many of the USPs, including USP7, USP4, USP1, etc., and there lie more opportunities for new inhibitor discovery.

- We agree that current inhibitors would not make use of these variations, but we feel that our findings could spark an interest in developing new classes that would benefit from the variability. We have adjusted the discussion to make that point more explicitly: "The variety in catalytic mechanisms might allow for development of new types of inhibitors with improved specificities."

- Similarly, it is a fancy term to cite of DUBTACs, but I don't see much relevance of this alternative residue applied to DUBTACs. The authors could explore the idea a bit if they decide to cite this.

- Indeed, only if the such new inhibitors can be made. We've removed the sentence on DUBTACS.

Minor comments and grammar: editing is difficult without the inclusion of line numbers. I have attempted to address errors the best I can, considering this.

- Synopsis: "..., the majority of USPs ****does**** not..." should be ****do****

- Correction was made

- Synopsis: "..., either critical ****residues**** can..." should be ****residue****

- Correction was made

- Intro: "Subsequently a tetrahedral..." should have a comma after subsequently

- Correction was made

- Intro: 2nd paragraph, line 6, be more specific to be "peptide bond."

- Correction was made

- Intro: in the 3rd paragraph, the residue numbers of the catalytic residues should be stated.

- The numbers were added

- Intro: the first line of paragraph 4. The statement is confusing and should be made clearer by simply stating, "The third catalytic residue in USPs is either Asp, Asn, or Ser."

- Correction was made

- Intro: second last paragraph, be a bit more specific on what "resembles USP15 and USP7" could be "... USP8, another USP whose catalytic triad resembles those of USP15 and USP7" because the domain structure of these FL USPs is very different, only the triad is similar.

- We agree and we apologize for this oversight, we have deleted the sentence on USP8 as it is not relevant in this context.

- Intro: the last paragraph mentions the loss of function USP15 mutation behaves like wild type and USP1. The term "loss of function" is misleading. If mutation to the canonical 3rd catalytic residue has no effect on activity, then it is not a loss of function mutant. Please specify the alanine mutation.

- We've made this change

- Intro: last paragraph, "Michaelis Menten," should have a hyphen in between.

- Correction was made

- Methods: please add a space between values and units; this comes up multiple times throughout the manuscript

- Corrections have been made

- Methods: all taxonomic names should be italicized, i.e., *E. coli*

- Correction was made

- Methods: protein stability section, "***build**-in" should be "***built**-in" (build-in is repeated elsewhere and needs to be fixed)

- Correction was made

- Methods: structure superposition section, "... bound to ubiquitin were **use** whenever..." should be "...bound to ubiquitin were **used** whenever..."

- Correction was made

- Methods: pH analysis section, "duplo" should be duplicate

- Correction was made

- Methods: Expression of USP1 in RPE1 cells section, please briefly state how you determined the expression level of USP1 in transduced RPE1 USP1KO cells when selecting clones with comparable levels to RPE1 wt cells

- We have added an extended description on how we selected these single clones. "To select clones with similar USP1 levels compared to endogenous, single clones were incubated with 1 µg/ml doxycycline for 44 hours and were lysed using RIPA buffer (1% NP40, 1% sodium deoxycholate, 0.1% SDS, 0.15 M NaCl, 0.01 M sodium

phosphate pH 7.5, 2 mM EDTA), containing cOmplete™, EDTA-free Protease Inhibitor Cocktail (Roche, 11873580001), 1 mM 2-chloroacetamide and 0.25 U/μl benzonase (SC-202391, Santa Cruz Biotechnology). Total protein concentration in the lysate was determined using a BCA assay (23227, Thermo Scientific) so that equal amounts could be loaded on gel. Samples were loaded on 4-12% Bolt gels (NW04127, Thermo Scientific), and run for 40 minutes at 180 V in MOPS running buffer (B0001, Thermo Scientific). Proteins were transferred to nitrocellulose membrane (10600002, Amersham Protran 0.45 NC nitrocellulose). Membranes were stained with a USP1 antibody (14346-1-AP, Proteintech). After incubation with HRP coupled secondary antibody the blots were imaged using a Bio-Rad Chemidoc XRS+. Using Bio-Rad ImageLab 5.1 software, USP1 levels were quantified by measuring the volume intensities of each USP1 band for each clone and compared this to endogenous USP1 levels in RPE1 cells. Clones with comparable expression levels were selected and used for further experiments.”

- Methods: tCoffee webserver should be "T-Coffee"

- We realized that multiple sequence alignment was performed using Clustal Omega, not T-Coffee, which has now been corrected. We apologize for this oversight.

- Methods: MSA. Can the authors provide more details on when doing BLAST, what were the criteria of selecting sequences from the result?

- Details have been added: “Catalytic domains as defined by Uniprot of the resulting human USPs were used for multiple sequence alignment. For USPs with multiple isoforms, the canonical isoform (isoform 1) was selected. In case of the USP17 gene family, USP17L2/DUB3 was selected (Komander et al., 2009). In order to properly align USP1, its inserts were removed from the catalytic domain following (Dharadhar et al., 2021). In order to properly align USP40, a shorter sequence was used (residues 250-480).”

- Methods: please provide the details for determining the concentration of the enzymes used.

- Details on how we determined the concentrations of enzymes have been added.

- Methods: Please provide the manufacturers of the Pherastar plate reader and the 384-well plate (please correct from "384 well-plate").

- Info on the manufacturers has been added.

- Results: In paragraph 1, "lies a ****much better**** conserved..." you should use "more highly."

- Correction was made

- Results: paragraph 1, "USP50 does not harbor either of" should be "USP50 harbors neither of"

- We corrected this: “This aspartate is present in all USPs except CYLD and USP50. The latter misses the third critical residue as well and therefore may be inactive.”

- Supp Fig 2: USP39 does not have glutamate in position of the first critical residue, it is glutamine (Q)

- Correction was made

- Results: second subsection title ****"The first critical residue is dispensable in USP1..."**** needs to be fixed

- Correction was made as follows: The third critical residue is dispensable in USP1/UAF1, USP15, USP40 and USP48

- Results: pg. 8 last line "to crosslink", the word crosslink is not proper for the reaction between Ub-PA with USPs. It usually refers to a reactive linker that links two molecules. Words like "conjugate", "conjugation," or "covalent react with", and "activity-based labelling" are probably better choices depending on the context.

- We have corrected this throughout the manuscript.

- Figure 1: figure legend describing B, C, and D are mixed up.

- Correction was made

- Results: In paragraph 9, the statement that your data on 5 USPs is representative of most of the 57 members in that the third catalytic is dispensable is not a sound statement for the small sample size. I think more emphasis on the diversity of USP1, USP7, USP15, USP40, and USP48 needs to be stated to help bolster such a claim. The statement to follow, which mentions sequence analysis alone is not able to predict the catalytic residue, is also somewhat contradictory to the opening statement and insinuates that all active USPs should be tested, while you only examined 5.

- We have changed this to "Our findings demonstrate that for the majority of tested USPs...". The diversity of tested USPs is clarified earlier in the manuscript: "These USPs vary in domain architecture and allosteric regulation, and therefore represent different aspects of the USP family, known for its structural variety and modular architecture". The statement about sequence analysis has been removed from the results section and is now only mentioned in the discussion. However, we do think that precise active site assignment for other USPs will require mutagenesis support.

- Figure 4: legend title, the critical residues are not responsible for **_performing_** nucleophilic attack per se; that is the job of Cys. The title of the figure should be altered to clear this up.

- Correction was made as follows: "Variation in the ability of USP critical residue mutants to successfully and efficiently facilitate a nucleophilic attack."

- Discussion: paragraph 3, since the Hu 2002 USP7 mechanism is not valid for other USPs tested, the "consensus USP catalytic mechanism" should be referred to as the "canonical."

- Indeed! Correction was made.

- Discussion: paragraph 4, "USP7, USP15 and USP40 all **three** have misaligned..." should be "USP7, USP15 and USP40 all have misaligned..."

- Correction was made.

- Discussion: paragraph 8, "negative charge itself could **contributes**..." should be "negative charge itself could **contribute**..."

- Correction was made

- Discussion: pg. 10, 3rd paragraph. Is the first sentence a statement of fact or a hypothesis? The writing is not clear to differentiate the two possibilities.

- Parts of the discussion have been rewritten, but the corresponding sentence has been rewritten as follows: "Canonically, it is thought that the fourth critical residue is involved in oxyanion hole formation."

- Discussion: pg. 10, 3rd paragraph, line 3, which "critical residue" does it refer to, the general base residue or the alternative residue?

- We've changed the text as follows: ". A dual role, with the third or fourth critical residue stabilizing catalytic histidine and oxyanion hole formation simultaneously is unlikely".

- Discussion: pg. 10, second last paragraph. Can the statement that "inaccurate assumptions about the catalytic triad ... be substantiated with an example?"

- We apologize for the possible confusion, but our point here was to point out that it could be misdirecting conclusions if you strictly follow the canonical assignment of the catalytic triad. We have rewritten the sentence to make that more clear: "Additionally, assumptions about the catalytic triad solely based on the canonical catalytic triad assignment in USP could affect conclusions made regarding loss of function mutations in

genetic screens. For example, we find that some USPs retain full or most of their activity once their canonical third catalytic residue is mutated.”

- Table 1, "ubiquitin variant" is mostly often used in the literature to refer to the ubiquitin mutants generated by phage display pioneered by the Sidhu lab or designed mutants. "ubiquitin and homolog derivatives" is a better term for "ubiquitin variant" in this article.

- We have changed this to ubiquitin-like proteins

- Table 1, the USP21 line "Lineair" is a typo, it should be "linear."

- Correction was made

- References: citations for Cadzow, 2020. and Tsefou, 2021 do not appear in the bibliography.

- Correction was made

- Add a hyphen to "Ubiquitin-specific proteases."

- Correction was made

Reviewer #2 (Significance (Required)):

General assessment:

Based on the studies of prototypical ubiquitin-specific protease USP7, the field generally accepts that USPs are a class of cysteine proteases that contain a catalytic triad with a cysteine, a histidine and a general base residue (asparagine, aspartate, or serine). This manuscript described the importance of an alternative, highly conserved aspartate that plays a critical role in catalysis using an enzyme kinetics study on five out of 57 USPs. The work is a very interesting observation that could change the perception in the field. However, the atomic details of how this fourth, or alternative residue, plays its role in catalysis are not clear without the structure evidence of an intermediate/transition state-bound complex.

Advance:

The study provided the first systematic enzymology study of the role of a fourth conserved residue critical for the catalysis of USPs. It is a conceptual advance and a first step to elucidate possibly a new catalytic mechanism of USPs.

Audience:

The manuscript will be of interest to biochemists in the field of ubiquitination and drug discovery.

Reviewers' expertise

The reviewers are structural biologists with expertise in the structure, function and enzymology of ubiquitin enzymes in general, with practical experience in drug discovery targeting the DUB and kinase families.

Reviewer #3 (Evidence, reproducibility and clarity (Required)):

The article by Keijzer and colleagues describes an interesting study comparing the active site of multiple USPs (the

largest subfamily of deubiquitinases) and elucidating the importance of specific residues lining the active site for catalysis. The authors carried out a careful analysis of the kinetic properties of 5 representative USPs and mutants thereof revealing a remarkable variety in their function that highlights that the majority of USPs studied do not require the canonical third residue of the catalytic triad of USPs for activity but instead rely on a highly conserved second critical residue. Furthermore, the authors apply complementary experimental approaches (mutagenesis, pH dependence of activity, crosslinking with Ub-PA) to allow distinguishing between residues important for the nucleophilic attack versus oxyanion hole stabilisation.

This is a well-written, thorough enzymatic study of high technical quality. The experiments are described in sufficient detail to allow others to reproduce the experimental set up. The data presented fully support the claims of the paper and no additional experiments are required to further support the conclusions.

It is great to see that the authors have carried out thermal stability assays on all WT and mutant proteins under investigation to ensure that any effects observed are not due to protein misfolding.

Minor comments:

- There are a few typos in the manuscript the authors should correct.

- Thank you, we have removed the typos from the manuscript.

- The panels/paper legends to Figure 1B/C/D are mixed up. Please correct.

- Correction was made

-It would be helpful to use different colours in the alignment shown in Supplementary Figure1 to indicate the position of the first and second critical residue.

- Thank you, we have highlighted these residues

- I wonder if the authors could comment on how representative the 5 USPs characterised in this work are of the entire family.

We address the variation of these USPs in more detail, both in the results as in the legend of figure 5: "These USPs vary in domain architecture and allosteric regulation, and therefore represent different aspects of the USP family, known for its structural variety and modular architecture"

Reviewer #3 (Significance (Required)):

Deubiquitinating enzymes (DUBs) play essential roles in many cellular processes and their activity is associated with a variety of diseases. There is a lot of interest in targeting DUBs for therapeutic purposes and a number of small molecule inhibitors are undergoing clinical studies.

While the structure and mechanism of multiple DUBs have been studied over the years, many open questions about their detailed catalytic mechanism remain and the importance of specific residues might often have been inferred based on sequence conservation alone without accompanying experimental support.

This work makes an important contribution to the field by systematically examining 5 members of the USP family and defining the precise role of the first and second critical residue for the catalytic cycle. This work will be of interest to those studying the mechanism of DUBs in general and those trying to target specific DUBs with small molecules. In addition, this study will also be interesting more generally for those studying enzyme kinetics as it highlights the importance of experimental validation of a catalytic mechanism that has been

January 8, 2024

RE: Life Science Alliance Manuscript #LSA-2023-02533

Prof. Titia K Sixma
Netherlands Cancer Institute
Biochemistry
Plesmanlaan 121
Amsterdam 1066 CX
Netherlands

Dear Dr. Sixma,

Thank you for submitting your revised manuscript entitled "Variety in the USP deubiquitinase catalytic mechanism". We would be happy to publish your paper in Life Science Alliance pending final revisions necessary to meet our formatting guidelines.

- please be sure that the authorship listing and order is correct
- please upload your main and supplementary figures as single files
- please add a Running Title and a Summary Blurb/Alternate Abstract to our system
- please add a Category for your manuscript in our system
- please add the Twitter handle of your host institute/organization as well as your own or/and one of the authors in our system
- please remove Graphical Abstract from the manuscript file and upload it separately with the file designation "Graphical Abstract.
- please consult our manuscript preparation guidelines <https://www.life-science-alliance.org/manuscript-prep> and make sure your manuscript sections are in the correct order
- please add a conflict of interest statement to your main manuscript text
- please add an Author Contributions section to your main manuscript text and the system
- please add your main, supplementary figure, and table legends to the main manuscript text after the references section
- please use the [10 author names et al.] format in your references (i.e., limit the author names to the first 10)
- please upload your Tables in editable .doc or Excel format
- please add callouts for Figures 2A, B and S2A to your main manuscript text

A. FINAL FILES:

B. MANUSCRIPT ORGANIZATION AND FORMATTING:

Sincerely,

Reviewer #1 (Comments to the Authors (Required)):

My initial concerns have been addressed in the revised manuscript and I now recommend publication.

Reviewer #2 (Comments to the Authors (Required)):

The authors have addressed all my previous concerns and I recommend the paper to be accepted.

January 23, 2024

RE: Life Science Alliance Manuscript #LSA-2023-02533R

Prof. Titia K Sixma
Netherlands Cancer Institute-Antoni van Leeuwenhoek Hospital
Biochemistry
Plesmanlaan 121
Amsterdam 1066 CX
Netherlands

Dear Dr. Sixma,

Thank you for submitting your Research Article entitled "Variety in the USP deubiquitinase catalytic mechanism". It is a pleasure to let you know that your manuscript is now accepted for publication in Life Science Alliance. Congratulations on this interesting work.

DISTRIBUTION OF MATERIALS:

Again, congratulations on a very nice paper. I hope you found the review process to be constructive and are pleased with how the manuscript was handled editorially. We look forward to future exciting submissions from your lab.

Sincerely,
